# FINCON: A Synthesized LLM Multi-Agent System with Conceptual Verbal Reinforcement for Enhanced Financial Decision Making

Yangyang Yu[1,*], Zhiyuan Yao[1,*], Haohang Li[1,*], Zhiyang Deng[1,*], Yuechen Jiang[1,*], Yupeng Cao[1,*]
Zhi Chen[1,*], Jordan W. Suchow[1], Zhenyu Cui[1], Rong Liu[1], Zhaozhuo Xu[1], Denghui Zhang[1]
Koduvayur Subbalakshmi[1], Guojun Xiong[2], Yueru He[3], Jimin Huang[3], Dong Li[3], Qianqian Xie[3,†]

[1]Stevens Institute of Technology    [2]Harvard University    [3]The Fin AI
*These authors contributed equally    † Corresponding author: `qianqian.xie@yale.edu`

## Abstract

Large language models (LLMs) have shown potential in complex financial tasks, but sequential financial decision-making remains challenging due to the volatile environment and the need for intelligent risk management. While LLM-based agent systems have achieved impressive returns, optimizing multi-source information synthesis and decision-making through timely experience refinement is underexplored. We introduce FINCON, an LLM-based multi-agent framework with **CON**ceptual verbal reinforcement for diverse **FIN**ancial tasks. Inspired by real-world investment firm structures, FINCON employs a **manager-analyst hierarchy**, enabling synchronized cross-functional agent collaboration towards unified goals via natural language interactions. Its **dual-level risk-control component** enhances decision-making by monitoring daily market risk and updating systematic investment beliefs through self-critique. These **conceptualized beliefs** provide verbal reinforcement for future decisions, selectively propagated to relevant agents, improving performance while reducing unnecessary peer-to-peer communication costs. FINCON generalizes well across tasks, including single stock trading and portfolio management. [1]

## 1 Introduction

The intricacies and fluctuations inherent in financial markets pose significant challenges for making high-quality, sequential investment decisions. In tasks such as single stock trading and portfolio management, each intelligent decision is driven by multiple market interactions and the integration of diverse information streams, characterized by varying levels of timeliness and modalities [1, 2]. The primary objective of these tasks is to maximize profit while managing present market risks in an open-ended environment.

In practice, trading firms often depend on synthesized teamwork, structured hierarchically with functional roles such as data analysts, risk analysts, and portfolio managers communicating across levels [3, 4]. These roles are responsible for the careful integration of diverse resources. However, the cognitive limitations of human team members can hinder their capacity to rapidly process market signals and achieve optimal investment outcomes [5].

To enhance investment returns and address the limitations of human decision-making, various studies have explored methods such as deep reinforcement learning (DRL) to develop agent systems that simulate market environments and automate investment strategies [6, 7, 8]. Concurrently,

---

[1]We will release the code and demo in the following repo `https://github.com/The-FinAI/FinCon`

38th Conference on Neural Information Processing Systems (NeurIPS 2024).

advancements in large language models (LLMs) have shown great potential in performing complex tasks, including reasoning [9, 10], tool-using [11], planning [12], decision-making [13, 14], and even in various financial applications [15, 16, 17, 18, 19], suggesting they may surpass existing agent architectures. Language agents, in particular, are distinguished by their human-like communication and flexible, prompt-based structures, making them well-suited to diverse decision-making settings [20, 21, 22, 23].

To achieve optimal decision-making performance, two critical factors must be considered: (1) Organizing agents to facilitate effective teamwork and efficient communication, and (2) Enabling agents to continuously learn and refine their actions. Studies have shown that mimicking human organizational structures can successfully coordinate language agents for specific tasks [24, 25, 26]. Additionally, recent advances in textual gradient-based prompt optimization [27, 28] and verbal reinforcement [29, 30] have proven effective in iteratively improving the reasoning and decision-making capabilities of language agents.

Language agent systems designed for financial decision-making, such as FINGPT [31], FINMEM [32], and FINAGENT [33], have shown strong performance. However, they face several limitations. First, their reliance on agents' risk preferences based on short-term market fluctuations fails to control long-term risk exposure, potentially overlooking fundamental factors driving investment returns. A more effective approach is to quantify investment risks using established **measures of risk** from quantitative finance [34, 35]. Second, these systems are often limited to single-asset trading tasks, making them less adaptable to multi-asset financial applications like portfolio management. Third, they place significant pressure on a single agent to understand and process information within a constrained context window, which can degrade decision quality. Although approaches like STOCKAGENT [36] use multi-agent systems for stock trading, their reliance on extensive discussions between numerous LLM agents leads to high communication costs and slow decision-making. Moreover, the absence of a clear optimization objective can compromise outcome effectiveness. Additional related work in the literature is discussed in the Appendix A.1.

To address these issues, we propose FINCON, an LLM-based multi-agent framework for critical financial tasks, such as single-stock trading and portfolio management, as shown in Figure 1. Our main contributions are: **1)** Inspired by real-world investment roles, we introduce a novel **Synthesized Manager-Analyst hierarchical communication structure** with a risk-control component. This structure allocates financial data from different sources to corresponding functional analyst agents, allowing them to focus on specific insights, while the manager consolidates these inputs to make informed trading decisions. The streamlined communication reduces redundant peer-to-peer interaction, lowering costs and improving efficiency. **2)** Our framework generalizes beyond stock trading to handle **portfolio management**, an area not previously addressed by other financial language agent systems. **3)** We developed a **dual-level risk control component** to update risk assessments both within and across episodes. Within episodes, risk is supervised using the Conditional Value at Risk (CVaR), a quantile-based risk measure [37]. Across episodes, we introduced a **verbal reinforcement** mechanism, where investment beliefs are updated based on reasoning trajectories and profit-and-loss (PnL) trends, distilled into **conceptual perspectives**. These insights are selectively back-propagated from the manager to relevant analyst agents. Our ablation studies demonstrate the effectiveness of this risk control design in managing market risk and enhancing trading performance.

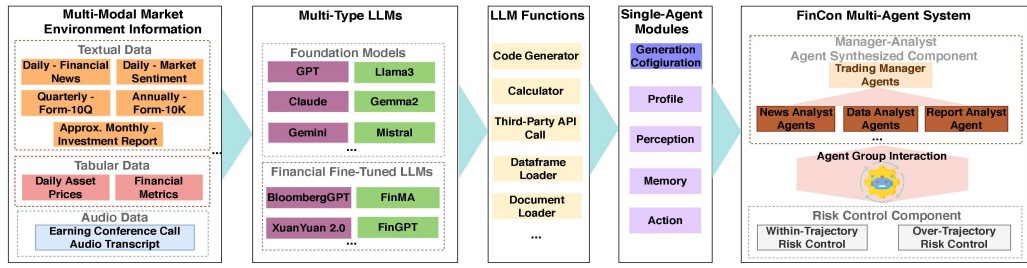

Figure 1: The general framework of FINCON.

## 2 Preliminaries

Here, we outline the mathematical notations for the two major financial decision-making tasks that will be explicitly discussed in our work. We also formally present the generalized modeling formulation using a Partially Observable Markov Decision Process (**POMDP**) [38] for financial decision-making tasks.

### 2.1 Financial Decision-making Tasks Formulation

**Single Stock Trading Tasks**. FINCON uses analyst agents group $\{M_{pr}^i\}_{i=1}^I$ to process multi-modal market information sources. The processed information is then used by a manager agent $M_a$ to make trading decisions (buy, sell, hold), and to provide relevant reasoning texts. Note that the "sell" signal means the system makes a "short-selling" decision, that is, a negative trading position is allowed. Additionally, FINCON evaluates the daily investment risk, followed by prompt-optimization for the manager agent from risk-control component $M_r$.

**Portfolio Trading Tasks**. In addition to processing multi-modal market information, the analyst agents also construct a stock pool for portfolio management by considering the statistical correlations between stock returns. The manager agent then makes trading decisions for each stock in the pool. Finally, the manager agent determines the portfolio weights for all stocks using an external optimization solver that applies the mean-variance optimization described below [39]:

$$\max_{\mathbf{w}}\langle\mathbf{w},\mu\rangle - \langle\mathbf{w},\Sigma\mathbf{w}\rangle \quad \text{s.t. } w_n = \begin{cases} \in [0,1], & \text{"buy"} \\ \in [-1,0], & \text{"sell"} \\ = 0, & \text{"hold"} \end{cases}, \quad \forall n \in \{1,\cdots,N\} \tag{1}$$

where $\mathbf{w} = (w_1\cdots,w_N) \in \mathbb{R}^N$ is portfolio weights vector, $\mu$ and $\Sigma$ are the shrinkage estimators of $N$-dimensional sample expected return and $N \times N$ sample covariance matrix of chosen stocks' daily return sequences respectively [35]. We note that portfolio weights are rebalanced on daily basis. In our implementation, we begin by calculating the portfolio weights through solving the aforementioned optimization problem. Next, the target positions are determined by linearly scaling these portfolio weights from the previous step.

### 2.2 Modeling Quantitative Trading as POMDP

Formally, we model quantitative trading task as an infinite horizon POMDP [40, 41] with time index $\mathbb{T} = \{0,1,2,\cdots\}$ and discount factor $\alpha \in (0,1]$. The components of this model are as follows: (1) a state space $\mathcal{X} \times \mathcal{Y}$ where $\mathcal{X}$ is the observable component and $\mathcal{Y}$ is unobservable component of the financial market; (2) the action space of analyst agents group is $\mathcal{A} = \prod_{i=1}^I \mathcal{A}^i$, where $\mathcal{A}^i$ represents the collection of processed market information in textual format done by agent $i$ (total $I$ analyst agents), and for manager agent, its action space is $\mathbb{A}$, which is modeled as $\{$*"buy", "sell", "hold"*$\}$ for single stock trading task and as $(\{$*"buy", "sell", "hold"*$\} \times [-1,1])^{\otimes N}$ for portfolio management task among $N$-stocks; (3) the reward function $\mathcal{R}(o,b,a): \mathcal{X} \times \mathcal{Y} \times \mathbb{A} \rightarrow \mathbb{R}$ uses daily profit & loss (PnL) as the output; (4) the observation process $\{O_t\}_{t\in\mathbb{T}} \subseteq \mathcal{X}$ is an $I$-dimensional process, with the $i^{th}$ entry $\{O_t^i\}_{t\in\mathbb{T}}$ representing one type of uni-modal information flow solely processed by the analyst agent $i$; (5) the reflection process $\{B_t\}_{t\in\mathbb{T}} \subseteq \mathcal{Y}$ represents the manager agent's self-reflection, which is updated from $B_t$ to $B_{t+1}$ on daily basis [42]); (7) the processed information flow $\hat{O}_t = (\hat{O}_t^1,\cdots,\hat{O}_t^I) \in \mathcal{A}, \forall t \in \mathbb{T}$, which represents the information processing outputs from analyst agents group.

Then, our multi-agent system is supposed to learn the policies of all agents: the policies of analyst agents $\pi_{\theta^i}^i: \mathcal{X} \rightarrow \mathcal{A}^i, i \in \{1,\cdots,I\}$ (the ways to process information, i.e. $\hat{O}_t^i \sim \pi_{\theta^i}^i(\cdot|O_t^i)$), and the policy of manager agent $\pi_{\theta^a}: \mathcal{A} \times \mathcal{Y} \rightarrow \mathbb{A}$ (the ways to make trading decisions, i.e. $A_t \sim \pi_{\theta^a}(\cdot|\hat{O}_t, B_t)$) such that the system maximizes cumulative trading reward while controlling risk [43]. All policies $\Pi_{\boldsymbol{\theta}} = (\{\pi_{\theta^i}^i\}_{i=1}^I, \pi_{\theta^a})$ are parameterized by textual prompts $\boldsymbol{\theta} = (\{\theta^i\}_{i=1}^I, \theta^a)$. By updating prompts via the risk-control component $M_r$, the whole system optimizes policies $\Pi_{\boldsymbol{\theta}}$ in a verbal reinforcement manner. By denoting daily profit & loss (PnL) by $R_t^{\Pi_{\boldsymbol{\theta}}} = \mathcal{R}(O_t, B_t, A_t)$, the

optimization objective for the whole system can be written as:

$$\max_{\boldsymbol{\theta}} \mathbb{E}\Big[\sum_{t\in\mathbb{T}} \alpha^t R_t^{\Pi^{\boldsymbol{\theta}}}\Big] \tag{2}$$

is a risk-sensitive optimization problem that leverages textual gradient descent, fundamentally differing from DRL algorithms designed for POMDPs. Further details on the textual gradient descent approach are provided in the Appendix A.2.

## 3 Architecture of FINCON

In this section, we present the architecture of FINCON using a two-level hierarchy. First, we describe the hierarchical framework for coordinating the agents' synchronous work and communication. Then, we elaborate on the functionalities of each module that constitutes each agent in FINCON. Finally, we aim to elaborate on how FINCON solves the objective function expressed as Equation (2) through a verbal reinforcement approach.

### 3.1 Synthesized Multi-agent Hierarchical Structure Design

The agent system of FINCON consists of two main components: the Manager-Analyst Agent Group component and the Risk-Control component.

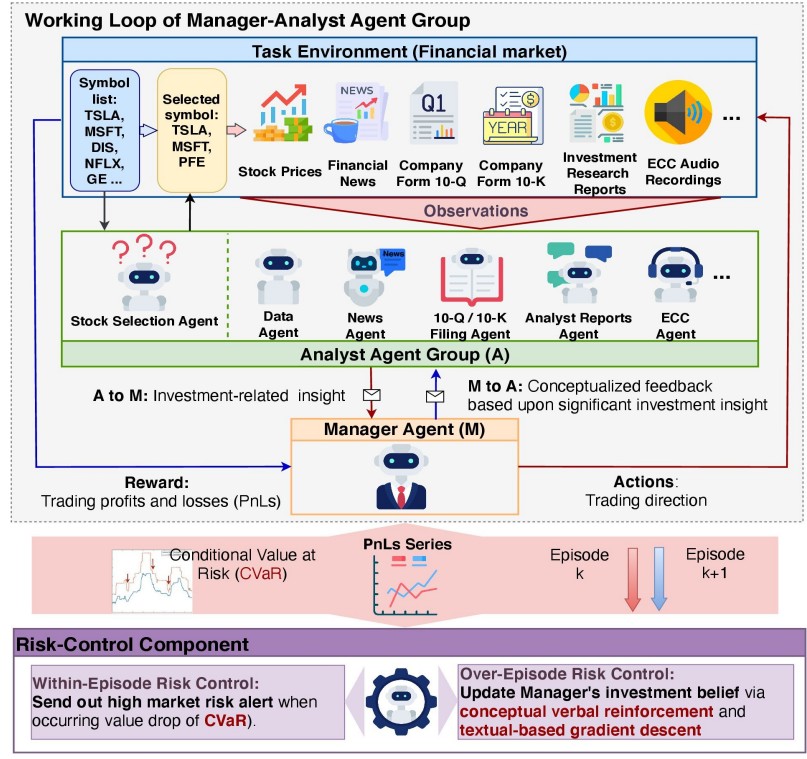

Figure 2: The detailed architecture of FINCON contains two key components: Manager-Analyst agent group and Risk Control. It also presents the between-component interaction of FINCON and decision-making flow.

### 3.1.1 Manager-Analyst Agent Group

Analogous to human investment firm, FINCON establishes a unique hierarchical structure to organize its multi-agent system, synthesizing their efforts to achieve superior decision-making outcomes.

The primary goal is to enhance information presentation and comprehension while minimizing unnecessary communication costs. The working mechanism of each agent is illustrated in Figure 2.

*Analyst Agents*. In FINCON, analyst agents distill concise investment insights from large volumes of multi-source market data, each focused on a specific trading target. To ensure high-quality reasoning by reducing task load and sharpening focus, each agent processes information from a single source in a uni-modal fashion, providing pre-specified outputs based on prompts. This setup mimics an efficient human team, where each analyst specializes in a specific function, filtering out market noise and extracting key insights. These agents assist the manager agent by consolidating denoised investment information from multiple perspectives. We implement seven distinct types of analyst agents using LLMs, each producing unique investment insights, as shown in the upper section of Figure 2. Based on input modalities, three textual data processing agents extract insights and sentiments from daily news and financial reports. An audio agent uses the Whisper API to interpret investment signals from earnings call recordings. Additionally, a data analysis agent and a stock selection agent compute critical financial metrics, such as momentum and CVaR, using tabular time series data. The stock selection agent also oversees portfolio selection by applying the classic risk diversification method in quantitative finance [1].

*Manager Agent*. In FINCON, the manager agent acts as the sole decision-maker, responsible for generating trading actions for sequential financial tasks. For portfolio management, it calculates portfolio weights using convex optimization techniques constrained by directional trading decisions (see optimization problem as presented in Formula (1)). Four key mechanisms support each decision: 1) Consolidating distilled insights from multiple analyst agents. 2) Receiving timely risk alerts and conceptual investment updates from the risk control component. 3) Refining its investment beliefs about the influence of different information sources on trading decisions for specific targets. 4) Conducting self-reflection by reviewing reasoning outcomes from previous trading actions.

### 3.1.2   Risk-Control Component

We have innovatively designed a dual-level risk-control mechanism consisting of within-episode and over-episode risk management. The within-episode mechanism detects market risk within a single training episode, allowing the manager agent to promptly adjust trading actions to mitigate potential losses by accounting for short-term trading performance and market fluctuations. This mechanism also operates during the testing phase. In contrast, the over-episode mechanism functions exclusively during the training stage, providing prompt optimization guidance by comparing the trading performance of the current episode with the previous one. This reflection enables the manager agent to update its investment beliefs based on performance differences. By drawing on prior observations of market risk and profitability patterns, these two mechanisms help avoid repeated investment errors, thereby enhancing future returns.

*Within-Episode Risk Control:* The within-episode risk alert is triggered by a sudden drop in the CVaR value. Conditional Value at Risk (**CVaR**) represents the average of the worst-performing 1% of daily trading Profits and Losses (**PnLs**). A decrease in CVaR typically indicates that recent trading decisions have led to PnLs within this bottom percentile, signaling a potentially high-risk market condition. When this occurs, the manager agent adopts a risk-averse stance for that day's trading actions, regardless of the prior risk status.

*Over-Episode Risk Control:* The over-episode investment belief updates facilitate adjustments in the emphasis placed on analysts' information distillation and the manager's action generation. Through the *Actor-Critic* mechanism, FINCON episodically optimizes its investment strategy for a given trading target, as defined by objective (Equation (2)), by reflecting on a series of winning and losing actions. This episodic reflection is powered by a unique *Conceptual Verbal Reinforcement (CVRF)*. CVRF assesses the performance of consecutive training episodes by analyzing the information perspectives provided by analysts and reflected in the manager's decision-making. It then conceptualizes and attributes the evaluation outcomes to these specific aspects. By comparing the conceptualized insights from more profitable versus less profitable episodes, the system informs both the manager and analyst agents about necessary belief adjustments, helping prioritize the most relevant market information for increased profitability, as detailed in Algorithm 1. CVRF leverages text-based gradient descent to offer optimal conceptual investment guidance for the manager agent, refining prompts with the latest investment beliefs. The guidance is organized according to perspectives provided by

the respective analyst agents, key financial indicators (such as historical momentum), or other crucial viewpoints.

| Factor | Gradient-based model optimizer | LLM-based prompt optimizer |
|---|---|---|
| Upgrade direction | Model value gradient momentum | Prompt reflection trajectory |
| Update method | Learning rate descent | Overlapping percentage of trading decisions |

Table 1: Analogy between glossaries in model optimizer and prompt optimizer.

These belief updates are first received by the manager agent and then selectively propagated to relevant agents, minimizing over-communication. Unlike the text-based gradient descent proposed by Tang et al.[28], which uses prompt editing distance as a learning rate, we derive investment belief updates by measuring the overlapping percentage of trading actions between two consecutive training trajectories at each belief update, as presented in Table 1. This approach has proven effective in improving the performance of a synthesized agent system, where each worker has a clearly defined and specialized role. The above describes the workflow of FINCON during the training stage, while the workflow during the testing stage is detailed in the Appendix A.3.

---

**ALGORITHM 1** Training Stage Algorithm of FINCON: Conceptual Verbal Reinforcement using Textual-based Gradient Descent

---

Initialize manager-analysts component $\{M_{pr}^i\}_{i=1}^I \& M_a$, and risk-control component $M_r$.
Initialize trading start date $s$, stock pool of portfolio and portfolio weights $w_0 = \mathbf{0}$.
Initialize Prompts $\boldsymbol{\theta}$, policy $\Pi_{\boldsymbol{\theta}}$.
**while** episode $k < Max$ **do**
    **for** $0 \leq t \leq T$ **do**
        Run policy $\Pi_{\boldsymbol{\theta}}$ (collecting daily PnL $r_t$, portfolio weights $w_t$ and daily CVaR value $\rho_t$).
        **if** $\rho_t < \rho_{t-1}$ **or** $r_t < 0$ **then**
            Trigger $M_a$ self-reflection and generate self-reflection text $B_t$.
        **end if**
        Get the investment trajectory $\mathcal{H}_k$ and calculate the objective function value (Function (2)).
    **end for**
    Compare the objective function values of episodes $k - 1 \& k$, and decide which episode has higher performance;
    Pass sustained profitable and losing trades from two episodes $\mathcal{H}_{k-1} \& \mathcal{H}_k$ into risk-control component $M_r$;
    Guide $M_r$ to summarize conceptualized investment insights $\{c_{k-1}^1, \cdots, c_{k-1}^n\} \& \{c_k^1, \cdots, c_k^m\}$;
    Compare two sets of conceptualized insights and give the reasoning for higher performance (providing textual optimization direction, i.e. $meta\ prompt$);
    Calculate the overlapping percentage between trading decision sequences from two episodes (providing the learning rate $\tau$);
    Update the prompts by textual gradient-descent:
$$\boldsymbol{\theta} \longleftarrow M_r(\boldsymbol{\theta}, \tau, meta\ prompt).$$
**end while**

---

## 3.2 Modular Design of FINCON Agents

Here, we explain the modular design of FINCON agents. Inspired by the recent works of Park et al. [44] and Sumers et al. [45] on developing the cognitive structure of language agents for human-like behavior, agents in FINCON integrate four modules to support their necessary functionalities, along with a shared general configuration, as detailed in Appendix A.4:

***General Configuration and Profiling Module***. This module defines task types (e.g., stock trading, portfolio management) and specifies trading targets, including sector and performance details. The profiling module outlines each agent's roles and responsibilities. The concatenated textual content from these parts is used to query investment-related events from the agents' memory databases. ***Perception Module***. This module defines how each agent interacts with the market, specifying the information they perceive, receive, and communicate, with interactions tailored to each agent's role. In detail, it converts raw market data, feedback from other agents, and information retrieved from the memory module into formats compatible with large language models, enabling them to process these inputs effectively. ***Memory Module***. The memory module comprises three key components:

*working memory*, *procedural memory*, and *episodic memory*. Much like how humans process events in their working memory [46], FINCON agents leverage their working memory to perform a range of tasks, including observation, distillation, and refinement of available memory events, all tailored to the specific roles of the agents. *Procedural memory* and *episodic memory* are critical for recording historical actions, outcomes, and reflections during sequential decision-making. Procedural memory is generated after each decision step within an episode, storing data as memory events. For trading inquiries, top events are retrieved from procedural memory and ranked based on recency, relevance, and importance, following a simplified version of the method proposed by Yu et al. [32], with further details provided in Appendix A.13. Each functional analyst agent has distinct procedural memory decay rates, reflecting the timeliness of various financial data sources, which is crucial for aligning multi-type data influencing specific time points and supporting informed decision-making. The manager agent enhances the procedural memory of analyst agents by providing feedback through an access counter. Both analyst and manager agents maintain procedural memory, but they keep different records, as illustrated in Appendix A.4. *Episodic memory*, exclusive to the manager agent, stores actions, PnL series from previous episodes, and updated conceptual investment beliefs from the risk control component.

## 4 Experiments

Our experiment answers the key research questions (RQs): **RQ1:** Does FINCON demonstrate robustness across multiple financial decision-making tasks, especially single-asset trading and portfolio management? **RQ2:** Is the within-episode risk control mechanism in FINCON effective in maintaining superior decision-making performance? **RQ3:** Is the over-episode risk control mechanism in FINCON effective in timely updating the manager agent's beliefs to enhance trading performance?

### 4.1 Experimental Setup

*(i) Multi-Modal Datasets.* We construct a market environment representation using real-world financial data, including stock prices, daily news, company filings (Form 10-Q, Form 10-K, etc.), and ECC audio from January 3, 2022, to June 10, 2023, as detailed in Appendix. A.8. Each data source is assigned to specific analyst agents based on its timeliness. *(ii) Evaluation Metrics.* We evaluate FINCON and other state-of-the-art (SOTA) agents using metrics such as Cumulative Return (CR%), Sharpe Ratio (SR), and Max Drawdown (MDD%). CR and SR are prioritized because they provide comprehensive insights into overall performance and risk-adjusted returns, essential for informed investment decisions. In contrast, MDD focuses on evaluating the potential for significant losses, making it a secondary consideration in this context. Details are provided in Appendix A.10. *(iii) Comparative Methods.* For single-stock trading, we compare FINCON with DRL agents (A2C, PPO, DQN) and LLM-based agents (GENERATIVE AGENT (GA), FINGPT, FINMEM, FINAGENT) as well as the Buy-and-Hold (B & H) strategy. For portfolio management, we compare FINCON with Markowitz MV, FinRL-A2C, and Equal-Weighted ETF strategy, with further details provided in Appendix A.12. The detailed experiment parameter configurations of the above agent systems are articulated in Appendix. A.14. *(iv) Implementation Details.* All LLM-based agents use GPT-4-Turbo, with temperature set at 0.3. FINCON is trained from January 3, 2022, to October 4, 2022, and tested from October 5, 2022, to June 10, 2023. DRL agents are trained over the period from January 1, 2018, to October 4, 2022, to ensure that there is sufficient data available for model convergence. Performance is based on the median CR and SR from five repeated epochs. For a more detailed explanation of the experimental setup, please refer to the Appendix A.5.

### 4.2 Main Results

In response to **RQ1**, we analyze FINCON's performance on two types of financial decision-making tasks: single-asset trading and portfolio management. The system's ability to manage these sequentially complex decisions is thoroughly evaluated in the following sections.

#### 4.2.1 Single Asset Trading Task

In this task, we evaluate FINCON's performance against other leading algorithmic trading models by trading eight different stocks. As presented in the tables above, FINCON significantly outperforms both LLM-based and DRL-based approaches in terms of CRs and SRs. Additionally, FINCON

achieves one of the lowest MDD values across most trading assets, demonstrating effective risk management while still delivering the highest investment returns. For detailed performance comparisons across all models and metrics, refer to Table 1.

Overall, even with extended training periods, DRL-based models tend to underperform, with the A2C algorithm lagging significantly behind other agents in general. Notably, the training periods for Nio Inc. (NIO) and Coinbase Global Inc. (COIN) require clarification. NIO, which completed its IPO in September 2018, has a slightly shorter training period than other tickers, yet the DRL algorithms for NIO still achieved convergence. In contrast, Coinbase Global Inc. (COIN), which completed its IPO in April 2021, presented a more significant challenge due to the limited available trading data, causing DRL algorithms to struggle with convergence. This limitation underscores a major drawback for DRL agents when trading recently listed IPOs. Consequently, our analysis of COIN focuses on comparisons between FINCON, LLM-based agents, and the buy-and-hold (B & H) strategy. In this context, FINCON demonstrates a clear advantage, achieving a cumulative return of over 57% and a Sharpe ratio of 0.825. Furthermore, LLM-based agents, which can leverage diverse data types and require minimal training, effectively mitigate the challenges faced by DRL algorithms.

| Categories | Models | TSLA | | | AMZN | | | NIO | | | MSFT | | |
|---|---|---|---|---|---|---|---|---|---|---|---|---|---|
| | | CR%↑ | SR↑ | MDD%↓ | CR %↑ | SR↑ | MDD%↓ | CR%↑ | SR↑ | MDD%↓ | CR%↑ | SR↑ | MDD%↓ |
| Market | B&H | 6.425 | 0.145 | 58.150 | 2.030 | 0.072 | 34.241 | -77.210 | -1.449 | 63.975 | 27.856 | 1.230 | 15.010 |
| Our Model | FINCON | 82.871 | 1.972 | 29.727 | 24.848 | 0.904 | 25.889 | 17.461 | 0.335 | 40.647 | 31.625 | 1.538 | 15.010 |
| LLM-based | GA | 16.535 | 0.391 | 54.131 | -5.631 | -0.199 | 37.213 | -3.176 | -1.574 | 3.155 | -31.821 | -1.414 | 39.808 |
| | FINGPT | 1.549 | 0.044 | 42.400 | -29.811 | -1.810 | 29.671 | -4.959 | -0.121 | 37.344 | 21.535 | 1.315 | 16.503 |
| | FINMEM | 34.624 | 1.552 | 15.674 | -18.011 | -0.773 | 36.825 | -48.437 | -1.180 | 64.144 | -22.036 | -1.247 | 29.435 |
| | FINAGENT | 11.960 | 0.271 | 55.734 | -24.588 | -1.493 | 33.074 | 0.933 | 0.051 | 19.181 | -27.534 | -1.247 | 39.544 |
| DRL-based | A2C | -35.644 | -0.805 | 61.502 | -12.560 | -0.444 | 37.106 | -91.910 | -1.728 | 68.911 | 21.397 | 0.962 | 21.458 |
| | PPO | 1.409 | 0.032 | 49.740 | 3.863 | 0.138 | 28.085 | -72.119 | -1.352 | 62.093 | -4.761 | -0.214 | 30.950 |
| | DQN | -1.296 | -0.029 | 58.150 | 11.171 | 0.398 | 31.174 | -35.419 | -0.662 | 56.905 | 27.021 | 1.216 | 21.458 |

| Categories | Models | AAPL | | | GOOG | | | NFLX | | | COIN | | |
|---|---|---|---|---|---|---|---|---|---|---|---|---|---|
| | | CR%↑ | SR↑ | MDD%↓ | CR %↑ | SR↑ | MDD%↓ | CR%↑ | SR↑ | MDD%↓ | CR%↑ | SR↑ | MDD%↓ |
| Market | B&H | 22.315 | 1.107 | 20.659 | 22.420 | 0.891 | 21.191 | 57.338 | 1.794 | 20.926 | -21.756 | -0.311 | 60.187 |
| Our Model | FINCON | 27.352 | 1.597 | 15.266 | 25.077 | 1.052 | 17.530 | 69.239 | 2.370 | 20.792 | 57.045 | 0.825 | 42.679 |
| LLM-based | GA | 5.694 | 0.372 | 14.161 | -1.515 | -0.192 | 8.210 | 41.770 | 1.485 | 20.926 | 19.271 | 0.277 | 67.532 |
| | FINGPT | 20.321 | 1.161 | 16.759 | 0.242 | 0.011 | 26.984 | 11.925 | 0.472 | 20.201 | -99.553 | -1.807 | 74.967 |
| | FINMEM | 12.397 | 0.994 | 11.268 | 0.311 | 0.018 | 21.503 | -10.306 | -0.478 | 27.692 | 0.811 | 0.017 | 50.390 |
| | FINAGENT | 20.757 | 1.041 | 19.896 | -7.440 | -1.024 | 10.360 | 61.303 | 1.960 | 20.926 | -5.971 | -0.106 | 56.882 |
| DRL-based | A2C | 13.781 | 0.683 | 14.226 | 8.562 | 0.340 | 21.191 | -8.176 | -0.258 | 49.579 | - | - | - |
| | PPO | 14.041 | 0.704 | 22.785 | 2.434 | 0.097 | 25.202 | -33.144 | -1.049 | 33.377 | - | - | - |
| | DQN | 21.125 | 1.048 | 16.131 | 20.690 | 0.822 | 21.191 | 21.753 | 0.687 | 39.733 | - | - | - |

Table 2: Comparison of key performance metrics during the testing period for the single-asset trading tasks involving eight stocks, between FINCON and other algorithmic agents. *Note that the highest and second highest CRs and SRs have been tested and found statistically significant using the Wilcoxon signed-rank test. The highest CRs and SRs are highlighted in red, while the second highest are marked in blue.*

In alignment with market trends, FINCON consistently exhibits superior decision-making quality compared to other LLM-based agents, regardless of market conditions—whether bullish (e.g., GOOG, MSFT), bearish (e.g., NIO), or mixed (e.g., TSLA). We attribute this performance to its high-quality distillation of information through a synthesized multi-agent collaboration mechanism, combined with its dual-level risk control design, positioning FINCON as a leader in the space. By contrast, FINGPT primarily relies on sentiment analysis of financial information, failing to fully exploit the potential of LLMs to integrate nuanced textual insights with numerical financial indicators. Similarly, GA and FINMEM use single-agent frameworks without sophisticated information distillation processes or a diverse toolset, placing heavy cognitive demand on the agent to process multi-source information, especially when dealing with large and varied data modalities. Moreover, their static or minimal investment belief systems result in weak filtering of market noise. As illustrated in Figure 7 (a) & (b) of Appendix A.7.2, this limitation leads these models to consistently hold lower positions and hesitate between 'buy' or 'sell' decisions, ultimately resulting in suboptimal performance.

FINCON overcomes these challenges through its innovative multi-agent synthesis, enabling it to deliver superior outcomes. Although FINAGENT performs well when integrating images and tabular data, it struggles to remain competitive when incorporating audio data, such as ECC recordings, which are critical in real-world trading. Additionally, FINAGENT relies on similarity-based memory retrieval, which can lead to decisions based on outdated information, often resulting in errors. In

contrast, FINCON's memory structure accounts for the varying timeliness of multi-source financial data, significantly enhancing decision quality and overall performance.

### 4.2.2 Portfolio Management Task

In this task, we compare FINCON's performance with the Markowitz Mean-Variance (MV) portfolio [47] and FINRL [48] in managing two small portfolios: Portfolio 1 (TSLA, MSFT, and PFE) and Portfolio 2 (AMZN, GM, and LLY). These assets were selected by the stock selection agent from a pool of 42 stocks, each with sufficient news data (over 800 news articles during the combined training and testing periods), as illustrated in Figure 9 in Appendix A.9. The training and testing periods, the backbone model and the parameter settings are consistent with those used in the single-asset trading task. For the Markowitz MV portfolio, we estimate the covariance matrix and expected returns using the same training data. In the case of FINRL, we use five years of training data prior to the test period. As detailed in Table 3 and Figure 3, our results show that FINCON outperforms both the Markowitz MV portfolio and FINRL as well as the market baseline – Equal-Weighted ETF, achieving significantly higher CRs and SRs, as well as MDDs.

However, managing multi-asset portfolios introduces more complexity, leading to a higher likelihood of hallucination compared to single-asset trading. This is due to the increased input length and complexity involved in multi-asset decision-making. While FINCON mitigates this issue by distributing tasks across specialized agents that focus on critical investment insights, it occasionally generates incorrect information, such as non-existent indices of memory events. Handling multi-asset decision-making requires sophisticated logic and substantial market information, which poses a significant challenge for LLMs when processing extended contexts. This complexity has left portfolio management relatively unexplored in previous language agent studies. Nonetheless, FINCON demonstrates considerable potential by constructing agent systems that can tackle complex financial tasks through effective resource optimization, even when managing relatively compact portfolios.

| Models | CR % ↑ | SR↑ | MDD %↓ | Models | CR % ↑ | SR↑ | MDD %↓ |
|---|---|---|---|---|---|---|---|
| FINCON | 113.836 | 3.269 | 16.163 | FINCON | 32.922 | 1.371 | 21.502 |
| Markowitz MV | 12.636 | 0.614 | 17.842 | Markowitz MV | 10.289 | 0.540 | 25.099 |
| FINRL-A2C | 19.461 | 0.831 | 26.917 | FINRL-A2C | 11.589 | 0.649 | 15.787 |
| Equal-Weighted ETF | 9.344 | 0.492 | 21.223 | Equal-Weighted ETF | 15.061 | 0.867 | 14.662 |

Table 3: Key performance metrics comparison among all portfolio management strategies of Portfolio 1 & 2. FINCON leads all performance metrics.

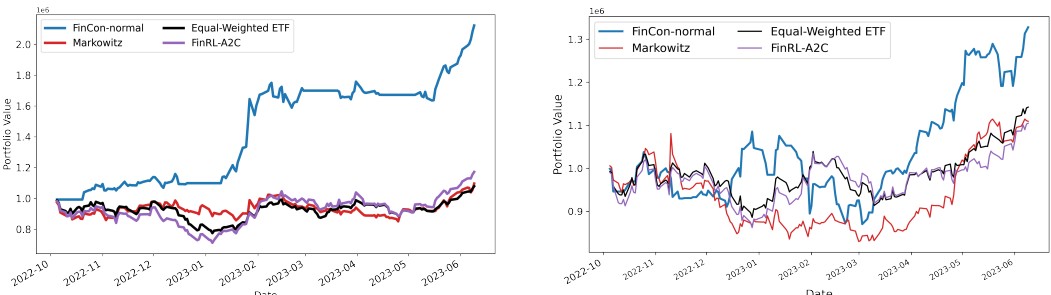

Figure 3: Portfolio values of Portfolio 1 & 2 changes over time for all the strategies. The computation of portfolio value refers to Equation 7 in Appendix A.10.

### 4.3 Ablation Studies

In response to **RQ2** and **RQ3**, we conduct a comprehensive evaluation of our unique risk control component through two ablation studies. Both studies maintain consistency with the training and testing periods used in the main experiments. The first study examines the effectiveness of the within-episode risk control mechanism, which leverages Conditional Value at Risk (CVaR) to manage risk in real-time, as detailed in Table 4. Comparisons on primary metrics illustrate that the success of utilizing CVaR for within-episode risk control is evident in both bullish and bearish market environments in the

single asset trading case. Moreover, in portfolio trading with mixed price trends, our within-episode risk control mechanism performs robustly by monitoring the entire portfolio's value fluctuations. The second study focuses on the over-episode risk control mechanism, demonstrating its critical role in updating the trading manager agent's beliefs to provide a more comprehensive understanding of current trading conditions, as articulated in Table 5. The markedly improved CRs and SRs in both decision-making scenarios underscore the effectiveness of using CVRF to update investment beliefs episodically, guiding the agent towards more profitable investment strategies. Additionally, FINCON demonstrates significant learning gains, achieving these results after only four training episodes—substantially fewer than what is typically required by traditional RL algorithmic trading agents. More visualizations and analysis are provided in the Appendix A.6.

| Task | Assets | Market Trend | Models | CR %↑ | SR↑ | MDD %↓ |
|------|--------|--------------|--------|-------|-----|--------|
| Single Stock | GOOG | General Bullish ↗ | w/ CVaR | 25.077 | 1.052 | 17.530 |
| | | | w/o CVaR | -1.461 | -0.006 | 27.079 |
| | NIO | General Bearish ↘ | w/ CVaR | 17.461 | 0.335 | 40.647 |
| | | | w/o CVaR | -52.887 | -1.002 | 70.243 |
| Portfolio Management | (TSLA, MSFT, PFE) | Mixed | w/ CVaR | 113.836 | 3.269 | 16.163 |
| | | | w/o CVaR | 14.699 | 1.142 | 17.511 |

Table 4: Key metrics FINCON with vs. without implementing **CVaR for within-episode risk control.** The performance of FINCON with the implementation of CVaR won a leading performance in both single-asset trading and portfolio management tasks.

| Task | Assets | Market Trend | Models | CR %↑ | SR↑ | MDD %↓ |
|------|--------|--------------|--------|-------|-----|--------|
| Single Stock | GOOG | General Bullish ↗ | w/ belief | 25.077 | 1.052 | 17.530 |
| | | | w/o belief | -11.944 | -0.496 | 29.309 |
| | NIO | General Bearish ↘ | w/ belief | 17.461 | 0.335 | 40.647 |
| | | | w/o belief | 8.197 | 0.156 | 55.688 |
| Portfolio Management | (TSLA, MSFT, PFE) | Mixed | w/ belief | 113.836 | 3.269 | 16.163 |
| | | | w/o belief | 28.432 | 1.181 | 27.535 |

Table 5: Key metrics FINCON with vs. without implementing **belief updates for over-episode risk control.** The performance of FINCON with the implementation of CVRF won a leading performance in both single-asset trading and portfolio management tasks.

## 5   Conclusion

In this paper, we present FINCON, a novel LLM-based multi-agent framework for financial decision-making tasks, including single stock trading and portfolio management. Central to FINCON is the Synthesized Manager-Analyst hierarchical communication structure and a dual-level risk control component. This communication method channels financial data from multiple sources to specialized analyst agents, who distill it into key investment insights. The manager agent then synthesizes these insights for decision-making. Our experimental evaluations demonstrate the efficacy of our risk control mechanism in mitigating investment risks and enhancing trading performance. Additionally, the streamlined communication structure reduces overhead. The dual-level risk control component introduces a novel approach to defining agent personas, enabling dynamic updates of risk and market beliefs within agent communication. A valuable future research direction would be to scale FINCON's framework to manage large-sized portfolios comprising tens of assets, while maintaining the impressive decision-making quality demonstrated with smaller portfolios. Given the LLM's input length constraint, a critical challenge lies in striking an optimal balance between information conciseness through agent distillation and potential performance deterioration when extending the current context window. Addressing this will be essential for ensuring quality-assured outcomes.

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

# A    Appendix

## A.1    Related Work

**LLM Agents for Financial Decision Making.** There are considerable efforts towards developing general-purpose LLM agent for sequential decision-making [49, 50], and such type of tasks often involve episodic interactions with environment and verbal reflections for action refinement, such as coding competition [51, 52], software development [53, 23], game-playing [54, 55]. Furthermore, researchers have started to exploit how LLM agents can perform better in harder decision-making tasks from finance [56, 57, 58, 59, 60], in which there are more volatile environments, leading to that the numerous unpredictable elements can obscure an agent's ability to reflect accurately on the reasons for poor decision outcomes. FinMem [32] enhances single stock trading performance by embedding memory modules with LLM agent for reflection-refinement, and FinAgent [33] improved trading profits via using external quantitative tool to fight against volatile environment.

**Multi-Agent System and Communication Structures.** In traditional multi-agent systems [61, 62], the way for agents' communication is pre-determined, like sharing data or state observations [63, 64, 65, 66, 67, 68]. The emergence of large language model brings flexibility for human-understandable communications [69, 20, 23, 70], so some work tries to elevate decision-making ability of LLM-based multi-agent system by letting agents engage in discussions [71, 21] or debates [72, 73]. The similar peer-communication strategy was as well utilized by the multi-agent system for financial tasks [74, 75, 76]. However, such approach are not optimal for unified-goal financial tasks that prioritize profits [77], because they suffer from potentially ambiguous optimization objectives and are unable to control the unnecessary communication costs [78].

**Prompt Optimization and Verbal Reinforcement.** To enhance the reasoning or decision-making of LLM agents, many prompt optimization techniques have been proposed, like ReAct [79], Chain of Thought (CoT) [80], Tree of Thoughts (ToT) [81], ART [14], intended for that LLM agents can automatically generate intermediate reasoning steps as an iterative program. In addition, to make LLM agents make decisions like humans and generate more understandable reasoning texts, some researchers recommend incorporating cognitive structures [82, 83, 44, 84]. Inspired by these previous work and DRL algorithms [85, 86, 87, 67, 88], verbal reinforcement [29, 30, 89, 24] was developed for LLM agents such that they can update actions based on iterative self-reflection while integrating additional LLM as a prompt optimizer [27, 28].

## A.2    Textual Gradient-Descent

In an LLM-based prompt optimizer, a meta-prompt [27, 28] is used to refine the task prompt for better performance. For example, for a mathematical reasoning task, the task prompt might be "Let's solve the problem," while the meta-prompt could be "Improve the prompt to help a model better perform mathematical reasoning."

Although prompt optimization lacks explicit gradients to control the update direction, we can simulate "textual gradient" by using LLMs' reflection capabilities. By generating feedback from past successes and failures on trading decisions, LLMs can produce "semantic" gradient signals that guide the optimization process.

Adjusting the optimization process's direction is crucial, similar to tuning the learning rate in traditional parameter optimization. An inappropriate learning rate can cause the process to oscillate or converge too slowly. Similarly, without proper control, the LLM-based optimizer might overshoot or oscillate during prompt optimization.

To mimic learning rate effects, we measure the overlapping percentage between trading decision sequences from consecutive iterations. We then directly edit the previous task prompt to enhance performance. The meta-prompt instructs the LLM to modify the current prompt based on feedback, ensuring a stable and incremental improvement process. This method allows for effective exploitation of existing prompts, leading to gradual performance enhancement.

## A.3 FINCON Testing Stage Workflow

During the testing stage, FINCON will utilize the investment beliefs learned from the training stage, and the over-episode risk control mechanism will no longer operate. However, the within-episode risk control mechanism will still function, allowing the manager agent to adjust trading actions in real-time based on short-term trading performance and market fluctuations. This ensures that even during testing, FINCON can promptly respond to market risks and potentially prevent losses while leveraging the knowledge gained during training.

---

**Algorithm 2** Testing Stage Algorithm of FINCON

---

Initialize trading start date $s$, stock pool of portfolio and portfolio weights $w_0 = \mathbf{0}$.
Inherit manager-analysts component $\{M_{pr}^i\}_{i=1}^I \& M_a$.
Inherit the reflections $B$, learned prompts $\boldsymbol{\theta}$, the trained policy $\Pi_{\boldsymbol{\theta}}$.
**for** $T + 1 \leq t \leq S$ **do**
  Run policy $\Pi_{\boldsymbol{\theta}}$ (collecting daily PnL $r_t$, portfolio weights $w_t$ and daily CVaR value $\rho_t$).
  **if** $\rho_t < \rho_{t-1}$ **or** $r_t < 0$ **then**
    Trigger $M_a$ self-reflection.
  **end if**
  Get one investment trajectory $\mathcal{H}$.
**end for**
Output performance metrics calculation results based on $\mathcal{H}$.

---

**A.4   Figure of Modular Design of Agents in** FINCON

General Configuration

1. Investment task introduction      2. Trading target background
3. Trading sectors                   4. Historical financial performance overview

Profiling Module

**Manager Agent:**
1. Role assignment: You are an experienced trading manager in the investment firm ...
2. Role description: Your responsibilities are to consolidate investment insights from analysts and make trading actions on {asset symbols} ...
**Analyst Agents:**
1. Role assignment: You are the investment analysts for news/ market data/ Form 10-K (Q)/ ECC audio recording ...
2. Role duty description: Your responsibilities are to distill investment insights and other indicators like financial sentiment for {asset symbols} ...

Perception Module

**Manager Agent:**
1. Perceive: Investment insights from analyst agents; daily risk alert and episode-level investment belief updates from the risk control component.
2. Send: Feedback to analyst agents about their contribution to significant investment earnings & losses.
**Analyst Agents:**
1. Perceive: Market information from certain information sources.
2. Send: Relevant market insights to manager agent.
3. Receive: Feedback from the manager agent.

Memory Module

**Manager Agent:**
1. Working:        - Consolidation       -Refinement        -Reflection
2. Procedural:     - Trading action records        - Reflection records
3. Episodic:       - Trajectory history
**Analyst Agents:**
1. Working:         - Observation        - Retrieval        - Distillation
2. Procedural:      - Distilled Investment-related insights    - Financial sentiment
                    - Investment report recommended actions

Action Module

**Manager Agent:**
1. Conduct: Trading actions.
2. Reflect: Trading reasons and analyst agents' contribution assessment.

Figure 4: The detailed modular design of the manager and analyst agents. The general configuration and profiling modules generate text-based queries to retrieve investment-related information from the agents' memory databases. The perceptual and memory modules interact with LLMs via prompts to extract key investment insights. The action module of the manager agent consolidates these insights to facilitate informed trading decisions.

## A.5    Experimental Setup

***Multi-Modal Datasets.*** We collect a comprehensive multi-modal dataset to simulate a realistic market environment. This dataset includes stock price data, daily financial news, company filing reports (10K and 10Q), and ECC (Earnings Call Conference) audio recordings spanning from January 3, 2022, to June 10, 2023. Each data source is assigned to specific analyst agents based on the timeliness of the information. For example, annual filings (10K) exhibit longer-term persistence, quarterly filings (10Q) and ECC data have medium-term relevance, and daily financial news provides the most immediate information.

***Evaluation Metrics.*** We evaluate FINCON and benchmark it against other state-of-the-art LLM-based and DRL-based agent systems using three key financial performance metrics: Cumulative Return (CR%) [90], Sharpe Ratio (SR) [91], and Max Drawdown (MDD%) [92]. These metrics help quantify each model's profitability, risk-adjusted returns, and risk management performance, respectively.

***Comparative Methods.*** In the single-stock trading task, we compare FINCON against seven algorithmic agents and the widely accepted Buy-and-Hold (B & H) baseline. The three DRL-based agents—A2C, PPO, and DQN—are from the FinRL framework [48], while the four state-of-the-art LLM-based agents include GENERATIVE AGENT [20], FINGPT [93], FINMEM [32], and FINAGENT [33]. For portfolio management, we benchmark FINCON against the classical Markowitz MV portfolio selection strategy [1], the RL-based FinRL-A2C agent [48], and the B & H strategy, which holds an equal-weighted position across all assets (equal-weighted ETF). Our focus on classical, RL-based, and B & H methods is due to the current lack of mature LLM-based agents for portfolio management tasks.

***Implementation Details.*** In our experiments, all LLM-based agent systems, including FINCON, use GPT-4-Turbo as the backbone model, with the temperature parameter set at 0.3 to balance response consistency with creative reasoning. FINCON is trained on financial data from January 3, 2022, to October 4, 2022, and tested on data from October 5, 2022, to June 10, 2023. Since deep reinforcement learning (DRL) agents require extensive data for convergence, their training period is extended to nearly five years (January 1, 2018, to October 4, 2022) to ensure fair comparison. The testing period remains the same across all models. The final performance metrics are based on the test trajectory with the median CR and SR values from five repeated epochs. If the median CR and SR occur in different epochs, performance is assessed based on the trajectory with the median CR value.

## A.6   Single Stock Trading Result Graphs

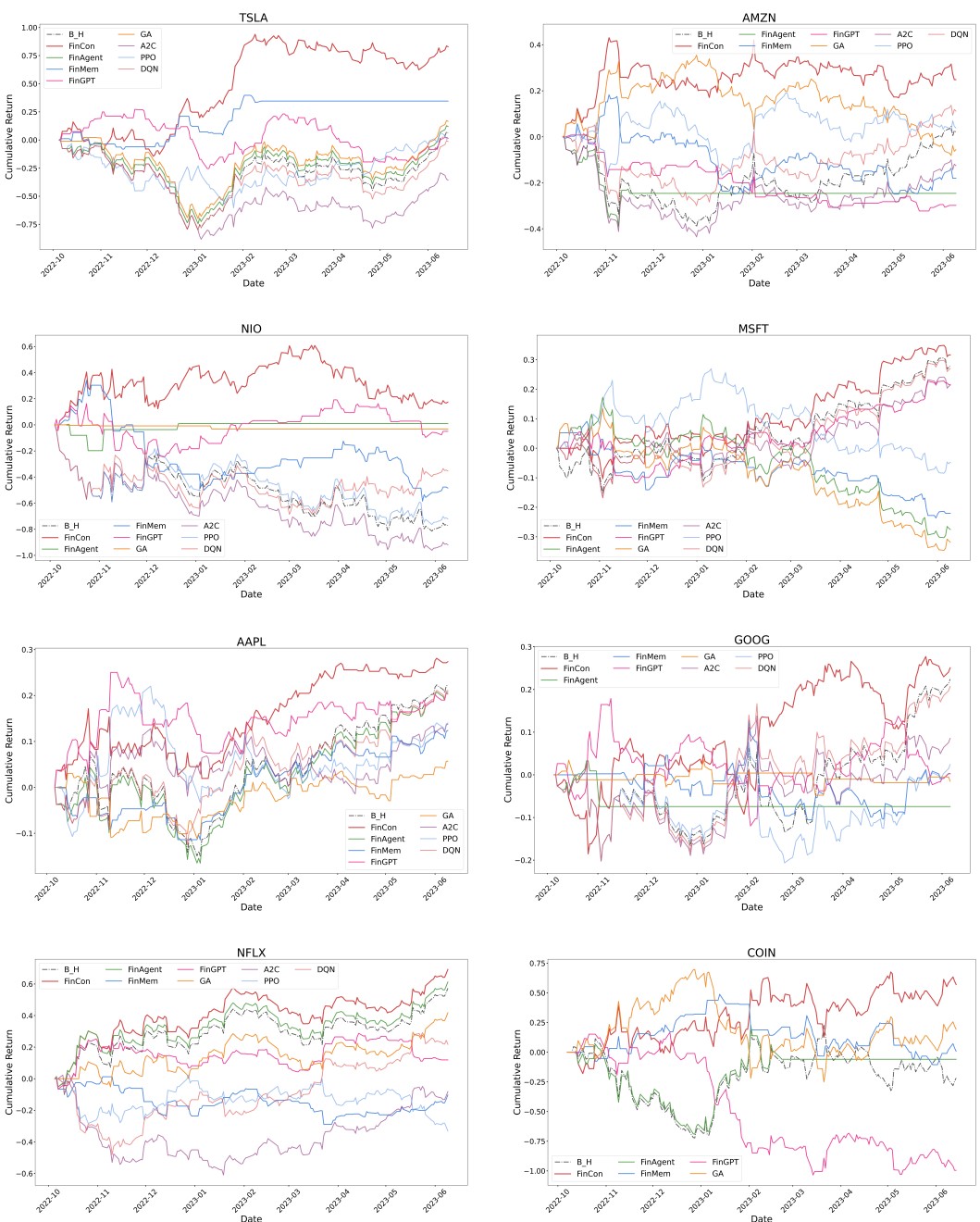

Figure 5: CRs over time for single-asset trading tasks. FINCON outperformed other comparative strategies, achieving the highest CRs across all six stocks by the end of the testing period, regardless of market conditions.

## A.7    Detailed Ablation Study

### A.7.1    The Effectiveness of Within-Episode Risk Control mechanism via CVaR

To answer the **RQ2**, we conduct the first ablation study. We assess the efficacy of FINCON's within-episode risk control mechanisms by monitoring system risk changes through CVaR. To demonstrate the robustness of FINCON, we compare the performance of FINCON with versus without CVaR implementation across two task types: single-asset trading and portfolio management. Furthermore, in single-asset trading tasks, we consider assets in both general bullish and bearish market conditions in the testing phase for comprehensive consideration.

Our results demonstrate that implementing CVaR in FINCON is highly effective across all financial metrics for both task types, as shown in Table 4 and Fig 6. For single-asset trading tasks, FINCON without within-episode risk control yields negative CRs and significantly higher MDDs, underperforming compared to the Buy-and-Hold strategy (CR of GOOG: 22.42%, CR of NIO: −77.210%), highlighting the severe consequences of ignoring environmental risks. In portfolio management, the CR increases dramatically from 14.699% to 113.836% with within-episode risk control, demonstrating its effectiveness in risk supervision even amid non-uniform market trends.

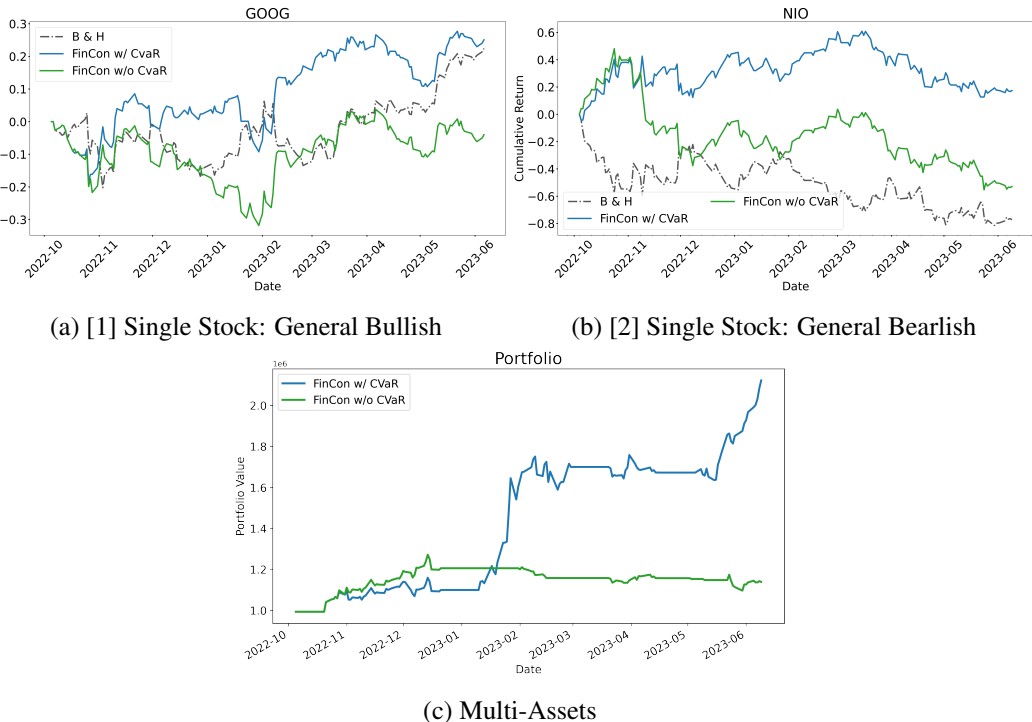

(a) [1] Single Stock: General Bullish          (b) [2] Single Stock: General Bearlish

(c) Multi-Assets

Figure 6: CRs of FINCON with vs. without implementing **CVaR for within-episode risk control** show that the CVaR mechanism significantly improves FINCON's performance. This is evident from two metrics: (a) cumulative returns over time for single stocks in both bullish and bearish market conditions, and (b) portfolio value over time for a multi-asset portfolio. In both cases, FINCON with CVaR demonstrates substantially higher gains.

Specifically, the success of utilizing CVaR for within-episode risk control is evident in both bullish and bearish market environments, as shown in the single asset trading case. In bullish markets, CVaR sharply captures immediate market shocks and timely informs FINCON to exercise caution, even amidst general optimism. Conversely, in bearish markets, CVaR consistently alerts FINCON to significant price drops, ensuring awareness of market risks. Moreover, in portfolio trading with mixed price trends, our within-episode risk control mechanism performs robustly by monitoring the entire portfolio's value fluctuations, enabling the trading manager agent to adjust potentially aggressive operations for each asset promptly.

### A.7.2    The Efficacy of Over-Episode Belief Updates Using CVRF

In the second ablation study, to answer **RQ3**, we use the same assets to examine the effectiveness of FINCON's over-episode risk control mechanisms. This is achieved by consistently improving FINCON's beliefs about market conditions for the targeted assets. To ensure consistent belief output for each training episode, we set the temperature parameter to 0 specifically for belief generation.

We collect third-time belief updates over four training episodes using our innovative CVRF mechanism. The overlap of trading actions between the last two adjacent episodes increases to over $80\%$, and the updated investment beliefs are mostly aligned. To illustrate FINCON's evolving investment beliefs through iterative training, we use the GOOG investment belief update as an example, as shown in Figure 8. Compared to the initial and final belief updates, each conceptual aspect, such as historical momentum and news insights, is enriched with executable information through our CRVF mechanism, leading to more profitable actions.

The results in Table 5 and Figure 7 indicate that the over-episode belief update mechanism is more critical than within-episode risk control in enhancing FINCON's decision-making. Without this functionality, key metrics like CR, SR, and MDD are lower than without the within-episode risk control in single asset trading. Although the CR of $28.432\%$ outperforms the Equal-Weighted ETF strategy's $9.344\%$ for portfolio management, the SR of $1.181$ is higher than Equal-Weighted ETF strategy's $0.492$, with the belief update feature, performance significantly further improves. It can achieve a CR of $113.836\%$ and an SR of $3.269$. These results demonstrate that using CVRF to update investment beliefs over episodes efficiently steers the agent's investment beliefs towards more profitable directions. FINCON achieves superior performance on multiple tasks, with learning gains evident after just four training episodes, requiring far fewer episodes than traditional RL algorithmic trading agents.

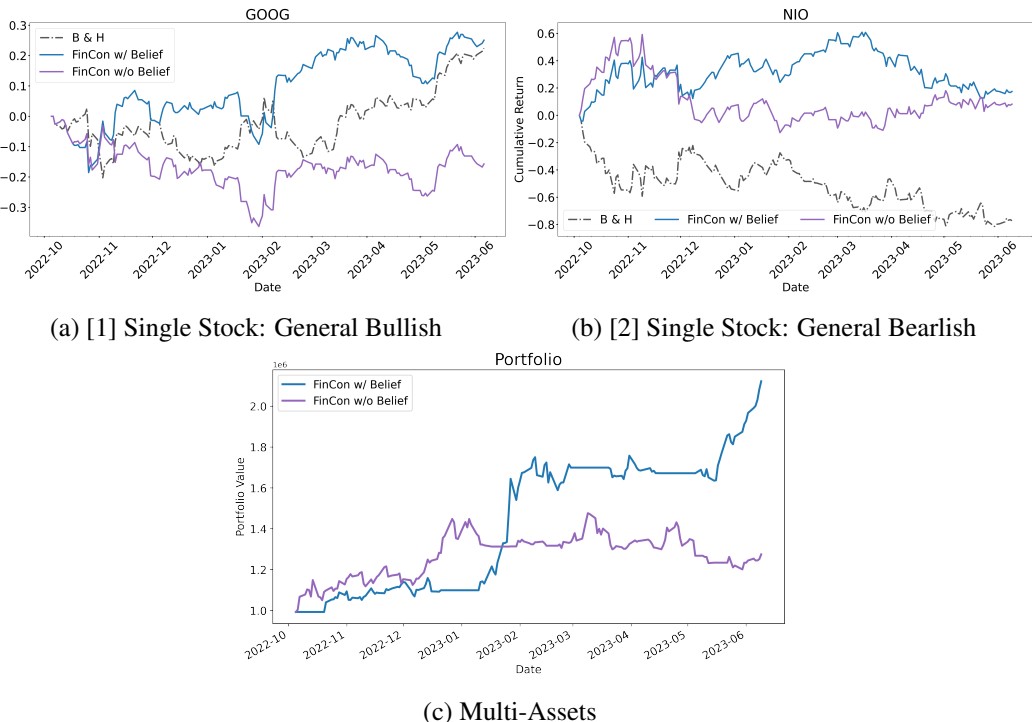

(a) [1] Single Stock: General Bullish                (b) [2] Single Stock: General Bearlish

(c) Multi-Assets

Figure 7: CRs of FINCON with vs. without **belief updates for over-episode risk control**. (a) The CRs over time for single stocks. The performance of FINCON with belief updates consistently leads in both bullish and bearish market conditions. (b) The portfolio values over time for multi-asset portfolio. FINCON's performance with belief updates also won a substantially higher gains.

**Updated Investment Belief Contents**

**First Update:**
{'historical momentum': 'Enhance the use of momentum indicators by establishing systematic rules for entry and exit based on momentum values to improve consistency and predictability in trading actions.',
'news insights': 'Integrate advanced real-time news sentiment analysis to better understand immediate market reactions and adjust trading strategies accordingly...',
'Form 10-Q': 'Incorporating annual report insights (10-K) could provide a comprehensive view of the company's long-term performance and strategic direction, aiding in more informed decision-making.'...,
'other aspects': ['sector trends', 'technological advancements'], ...}

**Last Update:**
{'historical momentum': 'The more profitable strategy effectively utilizes negative momentum values to make timely sell decisions, avoiding potential losses during downward trends, and effectively utilizes positive momentum values to make timely buy decisions, facilitating potential gains during upward trends. It is suggested to refine the use of momentum indicators, possibly by adjusting thresholds or incorporating additional short-term momentum metrics to enhance the timing and accuracy of buy/sell decisions.',
'news insights': 'It is recommended to further develop a systematic approach to quantify the impact of news, especially the sentiment scores and regulatory changes, to refine trading decisions.',
'Form 10-Q': 'It is suggested that even if there is a strong financial performance present 10-Q reports, it is better to prioritize immediate market signals. For future strategies, it is suggested to balance these aspects more evenly, especially in stable or bullish market conditions, to avoid premature exits from potentially profitable positions.',...,
'other aspects': ['sector trends', 'technological advancements', 'macroeconomic factors', 'regulatory risks'], ...}

---

**Tradng Action Overlapping Percentages**

1. Second vs. First Training Episode:  46.939%
2. Third vs. Second Training Episode: 71.429%
3. Fourth vs. Third Training Episode: 81.633%

Figure 8: The first time and last time LLM generated investment belief updates by CVRF for GOOG.

## A.8   Raw Data Sources

We assessed the performance of FINCON using multi-modal financial data from January 3, 2022, to June 10, 2022, sourced from reputable databases and APIs including Yahoo Finance (via yfinance), Alpaca News API, and Capital IQ, detailed explained in Table. These data, initially stored in the Raw Financial Data Warehouse as available observations of the financial market environment, are diverged into the corresponding FINCON's Analysts' Procedural Memory Databases based on timeliness through working memory's summarization operation.

| Data Sources |
|---|
| ***News data associated with ticker:*** News data is sourced from REFINITIV REAL-TIME NEWS mainly contains news from Reuters. |
| ***Form 10-Q, Part 1 Item 2 (Management's Discussion and Analysis of Financial Condition and Results of Operations):*** Quarterly reports (Form 10-Q) are required by the U.S. Securities and Exchange Commission (SEC). |
| ***Form 10-k, Section 7 (Management's Discussion and Analysis of Financial Condition and Results of Operations):*** Annual reports (Form 10-K) are required by the U.S. Securities and Exchange Commission (SEC), sourced from EDGAR, and downloaded via SEC API. |
| ***Historical stock price:*** Daily open price, high price, close price, adjusted close price, and volume data from Yahoo Finance. |
| ***Zacks Equity Research:*** |
| **Zacks Rank:** The Zacks Rank is a short-term rating system that is most effective over the one- to three-month holding horizon. The underlying driver for the quantitatively determined Zacks Rank is the same as the Zacks Recommendation and reflects trends in earnings estimate revisions. |
| **Zacks Analyst:** Reason to Sell, Reason to Buy, and potential risks. |
| ***Earning Conference Calls (ECC):*** ECC is a type of unstructured financial data (audio) that is crucial for understanding market dynamics and investor sentiment. The company executive board delivers ECC about recent financial outcomes, future projections, and strategic directions. Recent studies have underscored the importance of not only the textual content of these calls but also the audio feature. Analyses have revealed that the audio elements—such as tone, pace, and inflections—offer significant predictive value regarding company performance and stock movements [94, 95, 96]. |

Table 6: Raw data and memory warehouses of FINCON

## A.9   Distribution of Data

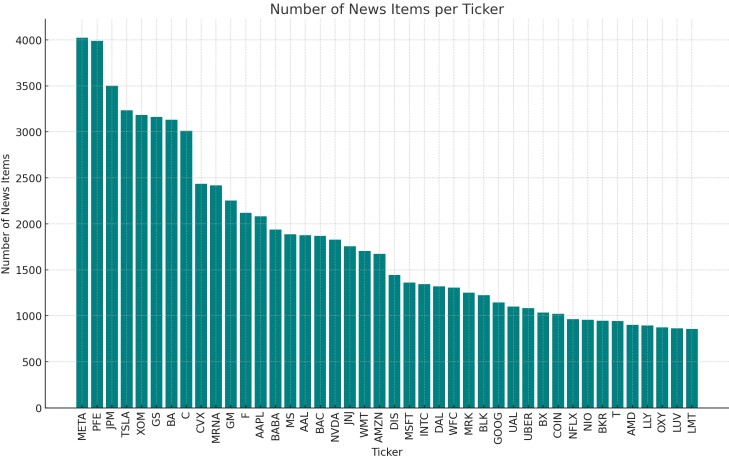

Figure 9: The distribution of news from REFINITIV REAL-TIME NEWS for the 42 stocks in the experiments

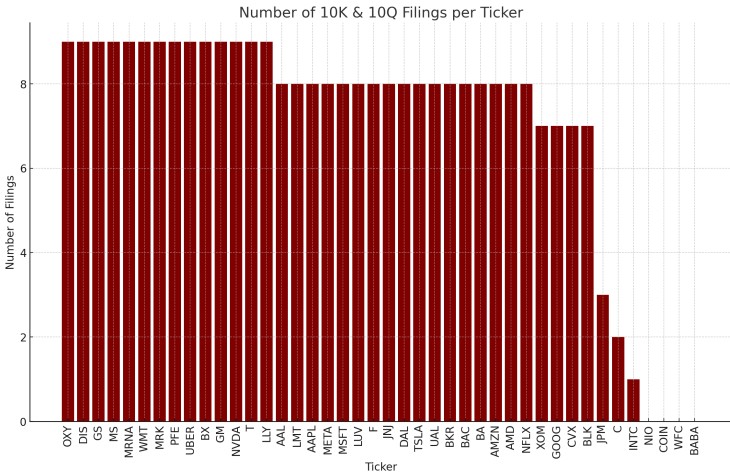

Figure 10: The distribution of 10k10q from Securities and Exchange Commission (SEC) for the 42 stocks in the experiments

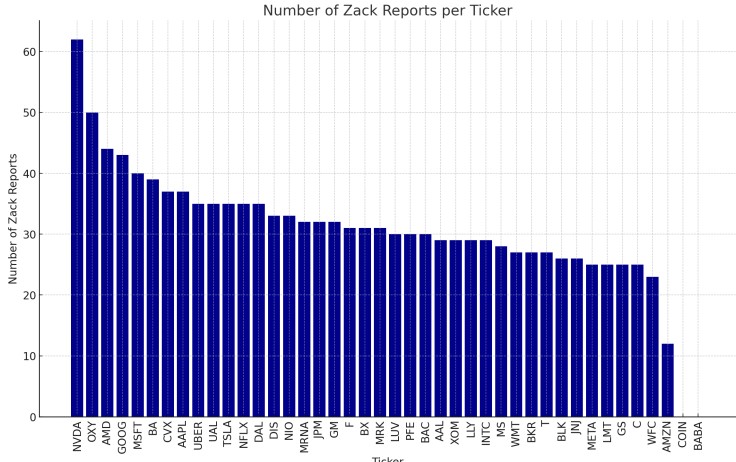

Figure 11: The distribution of Analyst Report from Zacks Equity Research for the 42 stocks in the experiments

### A.10 Formulas of Classic Financial Metrics for Risk Estimator and Decision-making Task Performance Evaluation

The risk estimator uses the following metrics:

**Profit and Loss (PnL)**[97]: PnL quantifies the net outcome of trading activities over a specified period by accounting for the realized gains and losses from financial instruments like stocks and derivatives.

**Value at Risk (VaR)** of PnL[97]: VaR is a statistical tool used to estimate the potential loss in a portfolio, within a defined confidence interval. Mathematically, it is defined as Equation 3:

$$\text{VaR}_\alpha(PnL) = \inf\{l \in \mathbb{R} : \mathbb{P}(PnL \le l) \ge \alpha\} \tag{3}$$

where $\alpha$ is the confidence level.

**Conditional Value at Risk (CVaR)** of PnL[97]: CVaR is a statistical tool used to estimate the expected potential loss worse than the VaR value in a portfolio, within a defined confidence interval. Mathematically, it is defined as Equation 4:

$$\text{CVaR}_\alpha(PnL) = \mathbb{E}\Big\{PnL | PnL \le \text{VaR}_\alpha(PnL)\Big\} \tag{4}$$

where $\alpha$ is the confidence level.

The performance evaluation of algorithmic trading agents incorporates the following metrics:

**Cumulative Return of PnL** [90]: Cumulative Return is a key trading performance metric because it provides a comprehensive insight into investment performance, especially for strategies that emphasize long-term growth and reinvestment. The effectiveness of different investment strategies is evaluated based on their Cumulative Returns, which reflect the total change in value over time. In this study, we compute Cumulative Returns over the specified period by summing daily logarithmic returns, as outlined in Equation 5. This method is widely accepted in the finance area due to its ability to precisely capture minor price fluctuations and symmetrically address gains and losses. In essence, a higher Cumulative Return typically indicates a more effective strategy.

$$
\begin{aligned}
\textbf{Cumulative Return} &= \sum_{t=1}^{n} r_i \\
&= \sum_{t=1}^{n} \left[ \ln\left( \frac{p_{t+1}}{p_t} \right) \cdot \text{action}_t \right],
\end{aligned} \tag{5}
$$

where $r_i$ represents the PnL for day $t + 1$, $p_t$ is the closing price on day $t$, $p_{t+1}$ is the closing price on day $t + 1$, and action$_t$ denotes the trading decision made by the model for that day.

**Portfolio Value**: Portfolio value represents the total worth of all the investments held in a portfolio at a given point in time. It is a metric used only in the portfolio management task.

$$
\textbf{Cumulative Simple Return}_t = \prod_{k=1}^{t} (1 + \textbf{Daily Simple Return}_t) - 1 \tag{6}
$$

$$
\textbf{Portfolio Value}_t = \textbf{Initial Investment Amount} \times (1 + \textbf{Cumulative Simple Return}_t) \tag{7}
$$

, where the initial amount is set as $\$1,000,000$.

**Sharpe Ratio of PnL** [91]: Sharpe Ratio is another core metric for evaluating investment performance and adjusting returns for risk. It is calculated by dividing the portfolio's average PnL ($R_p$) over the risk-free rate ($R_f$) by its volatility ($\sigma_p$), as shown in Equation 8. This metric adjusts returns for risk, with a higher ratio indicating better risk-adjusted performance. Essential in comparing different portfolios or strategies, it contextualizes performance against similar investments. Although a Sharpe Ratio above 1 is typically considered favorable and above 2 as excellent, these benchmarks can vary depending on the context of comparison.

$$
\textbf{Sharpe Ratio} = \frac{R_p - R_f}{\sigma_p} \tag{8}
$$

**Max Drawdown of PnL** [92]: Max Drawdown is a metric for assessing risk. It represents the most significant decrease in a portfolio's value, from its highest ($P_{\text{peak}}$) to its lowest point ($P_{\text{trough}}$) until a new peak emerges, detailed in Equation 9. Indicative of investment strategy robustness, a smaller Max Drawdown suggests reduced risk.

$$
\textbf{Max Drawdown} = \max\left( \frac{P_{\text{peak}} - P_{\text{trough}}}{P_{\text{peak}}} \right) \tag{9}
$$

## A.11 Baseline and Comparative Models on Single Stock Trading Task

**Buy-and-Hold strategy (B&H)**:

A passive investment approach, where an investor purchases stocks and holds onto them for an extended period regardless of market fluctuations, is commonly used as a baseline for comparison of stock trading strategies.

**LLM trading agents:**

We evaluate FINCON against four LLM agents in the context of stock trading.

- **GENERAL-PURPOSE GENERATIVE AGENTS – GA:** The generative AI agent by Park et al. [20], originally intended to simulate realistic human behavior and make everyday decisions, has been adapted here for specific stock trading tasks. This agent's architecture includes a memory module that employs recency, relevance, and importance metrics to extract pivotal memory events for informed decision-making. However, it does not provide a layered memory module to effectively differentiate the time sensitivities unique to various types of financial data. Additionally, although it features a profiling module to define agent attributes like professional background, the model does not specify the agent's persona. In our experiments, we modified the original prompt template created by Park et al., which was intended for general daily tasks, to suit financial investment tasks.
- **FINGPT:** A novel open-source LLM framework specialized for converting incoming textual and numeric information into informed financial decision-making, introduced by Yang et al[31]. It claims superiority over the traditional buy-and-hold strategy.
- **FINMEM:** FINMEM employs a specialized profiling module and self-adaptive risk settings for enhanced market robustness. Its memory module integrates working memory and layered long-term memory, enabling effective data processing. This allows FINMEM to leverage market insights and improve trading decisions [32].
- **FINAGENT:** FINAGENT developed upon FINMEM, which leverages the use of tool-using capabilities of LLMs to incorporate multi-modal financial data [33]. It claims an further improved trading performance on single asset trading (stocks and cryptocurrencies).

**DRL trading agents:**

As the FINMEM is practiced and examined on the basis of single stock trading and discrete trading actions, we choose three advanced DRL algorithms fitting into the same scenarios according to the previous and shown expressive performance in the work of Liu et al [98, 99]. The DRL training agents only take numeric features as inputs.

- **Advantage Actor-Critic (A2C):** A2C ([100]) is applied to optimize trading actions in the financial environment. It operates by simultaneously updating both the policy (actor) and the value (critic) functions, providing a balance between exploration and exploitation.
- **Proximal Policy Optimization (PPO):** PPO ([101]) is employed in stock trading due to its stability and efficiency. One salient advantage of PPO is that it maintains a balance between exploration and exploitation by bounding the policy update, preventing drastic policy changes.
- **Deep Q-Network (DQN):** DQN ([102]) is an adaptation of Q-learning, that can be used to optimize investment strategies. Unlike traditional Q-learning that relies on a tabular approach for storing Q-values, DQN generalizes Q-value estimation across states using deep learning, making it more scalable for complex trading environments.

## A.12   Portfolio Management

**Markowitz Portfolio Selection** [1]: introduced by Harry Markowitz in 1952, is a framework for constructing portfolios that optimize expected return for a given level of risk or minimize risk for a given level of expected return. This method uses expected returns, variances, and covariances of asset returns to determine the optimal asset allocation, thereby balancing risk and return through diversification.

**FinRL-A2C** [48]: is an RL algorithm proposed to address single stock trading and portfolio optimization problems in Liu et al.. The RL models make trading decisions (i.e., portfolio weights) based on the observation of previous market conditions and the brokerage information of the RL agents. The implementation of this algorithm [2] is provided and is used as baselines in our study.

**Equal-Weighted ETF** [103]: is a portfolio giving equal allocation to all stocks, similar to a buy-and-hold strategy in single-stock trading, can provide a benchmark on market trends.

---

[2] `https://github.com/AI4Finance-Foundation/FinRL-Meta`

## A.13 Ranking Metrics for Procedural Memory in FINCON

Upon receiving an investment inquiry, each agent in FINCON retrieves the top-$K$ pivotal memory events from its procedural memory, where $K$ is a hyperparameter. These events are selected based on their information retrieval score. For any given memory event $E$, its information retrieval score $\gamma^E$ is defined by

$$\gamma^E = S^E_{\text{Relevancy}} + S^E_{\text{Importance}} \tag{10}$$

which is adpated from Park et al [20] but with modified relevancy and importance computations, and is scaled to $[0, 1]$ before summing up. Upon the arrival of a trade inquiry $P$ in processing memory event $E$ via LLM prompts, the agent computes the relevancy score $S^E_{\text{Relevancy}}$ that measures the *cosine similarity* between the embedding vectors of the memory event textual content $\mathbf{m_E}$ and the LLM prompt query $\mathbf{m_P}$, which is defined as follows:

$$S^E_{\text{Relevancy}} = \frac{\mathbf{m_E} \cdot \mathbf{m_P}}{\|\mathbf{m_E}\|_2 \times \|\mathbf{m_P}\|_2} \tag{11}$$

Note that the LLM prompt query inputs trading inquiry and trader characteristics. On the other hand, the importance score $S^E_{\text{Importance}}$ is inversely correlates with the time gap between the inquiry and the event's memory timestamp $\delta t = t_{\text{P}} - t_E$, mirroring Ebbinghaus's forgetting curve [104]. More precisely, if we denote the initial score value of memory event $v^E$ and degrading ratio $\theta \in (0, 1)$, then the importance score is computed via

$$S^E_{\text{Importance}} = v^E \times \theta^{\delta t} \tag{12}$$

Note that the ratio $\theta$ measures the diminishing importance of an event over time, which is inspired by design of [20]. But in our design, the factors of recency and importance are handled by one equation. Different agents in FINCON admit different choices of $\{v^E, \theta\}$ for memory event $E$.

Additionally, an access counter function facilitates memory event augmentation, so that critical events impacting trading decisions can be augmented by FINCON, while trivial events are gradually faded. This is achieved by using the LLM validation tool Guardrails AI [105] to track critical memory ID. A memory ID deemed critical to investment gains receives $+5$ to its importance score $S^E_{\text{Importance}}$. This access counter implementation enables FINCON to capture and prioritize crucial events based on type and retrieval frequency.

## A.14 Detailed Configurations in Experiments

The training period was chosen to account for the seasonal nature of corporate financial reporting and the duration of data retention in FINCON's memory module. The selected training duration ensures the inclusion of at least one publication cycle of either Form 10-Q, ECC, or Form 10-K. This strategy ensures that the learned conceptualized investment guidance considers a more comprehensive scope of factors. Additionally, the training duration allowed FINCON sufficient time to establish inferential links between financial news, market indicators, and stock market trends, thereby accumulating substantial experience. Furthermore, we set the number of top memory events retrieved for each agent at 5. We ran FINCON.

To maintain consistency in the comparison, the training and testing phases for the other three LLM-based agents were aligned with those of FINMEM. For parameters of other LLM-based agents that are not encompassed by FINMEM's configuration, they were kept in accordance with their original settings as specified in their respective source codes.

FINCON's performance was benchmarked against that of the most effective comparative model, using Cumulative Return and Sharpe Ratio as the primary evaluation metrics. The statistical significance of FINCON's superior performance was ascertained through the non-parametric Wilcoxon signed-rank test, which is particularly apt for the non-Gaussian distributed data.

## A.15 FINCON performance on extreme market conditions

To further illustrate the robustness of FINCON's performance, we assess its effectiveness in two distinct scenarios: (1) a single-asset trading task using TSLA and (2) a portfolio management task involving a combination of TSLA, MSFT, and PFE. Our evaluation focuses on key financial metrics, including Cumulative Returns (CRs), Sharpe Ratios (SRs), and Maximum Drawdown (MDD). The

training period spanned from January 17, 2022, to March 31, 2022, while the testing phase covered April 1, 2022, to October 15, 2022. This specific timeframe was chosen due to the elevated levels of the CBOE Volatility Index (VIX), which averaged above 20, signaling greater market volatility during these months.

As demonstrated in Table 7 and Figure 12, FINCON is the sole agent system to achieve positive Cumulative Returns (CRs) and Sharpe Ratios (SRs) in single stock trading tasks. Regarding portfolio management tasks, the results of all baselines (four benchmarks) are detailed in Table 8 and Figure 13. In these comparisons, FINCON consistently attained the highest scores in the primary performance metrics.

| Models | CR % ↑ | SR↑ | MDD %↓ |
|---|---|---|---|
| B&H | -56.738 | -1.625 | 52.077 |
| FINCON | 22.460 | 0.695 | 45.215 |
| GA | -51.251 | -1.547 | 48.763 |
| FINGPT | -20.035 | -0.805 | 32.199 |
| FINMEM | -47.809 | -1.549 | 49.560 |
| FINAGENT | -31.119 | -1.933 | 33.224 |
| A2C | -73.251 | -2.142 | 56.998 |
| PPO | -78.007 | -2.284 | 59.003 |
| DQN | -8.452 | -1.328 | 8.463 |

Table 7: Key performance comparison for single asset trading under the high volatility condition using TSLA as an example. FINCON leads all performance metrics.

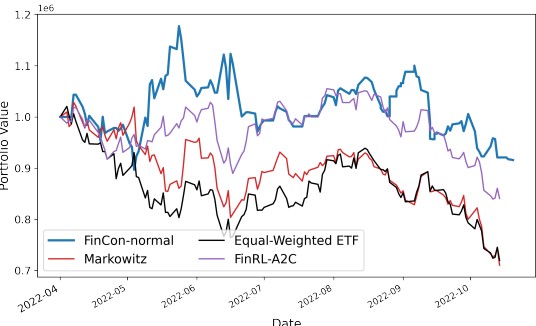

Figure 12: CR changes over time across all the strategies under the high volatility condition using TSLA as an example of the single asset trading task.

| Models | CR % ↑ | SR↑ | MDD %↓ |
|---|---|---|---|
| FINCON | -8.429 | -0.294 | 26.176 |
| Markowitz MV | -28.996 | -1.805 | 31.831 |
| FINRL-A2C | -15.932 | -1.195 | 21.569 |
| Equal-Weighted ETF | -28.008 | -1.731 | 30.070 |

Table 8: Key performance comparison among all portfolio management strategies of Portfolio1 under the high volatility condition. FINCON leads all performance metrics.

Figure 13: Portfolio1 value changes over time for all the strategies under the high volatility condition.

