# OpenReview forum: "FinCon: A Synthesized LLM Multi-Agent System with Conceptual Verbal Reinforcement for Enhanced Financial Decision Making"
_NeurIPS.cc/2024/Conference — NeurIPS 2024 poster_

### Official Review · Reviewer_pGXR · 2024-07-02

**Soundness:** 3
**Presentation:** 3
**Contribution:** 3
**Rating:** 7
**Confidence:** 3

**Summary:**

The paper proposes a specialized multi-LLM-agent composition for (1) stock trading and (2) portfolio management. The authors collect historical multimodal data up to 10 years to evaluate their setup. Inspired by the real world financial institution structure, the authors instruct the 8 specialized agents to perform stock and financial report analysis as well as final decision making. The authors introduce conceptual verbal reinforcement (CVR) to improve the initial prompts of the agents.

**Strengths:**

[1] The paper tries to apply LLM agents to the financial problems which is a very novel and promising field of study.

[2] The study is well structured and encloses the comparison with 6 state of the art competitors and the B&H baseline for single stock trading and 3 competitors for portfolio management. 3 different metrics are taken into account: CR%, SR and MDD% that helps to obtain a reliable signal about a method's performance. Specifically Figure 3 shows impressive results visible by a naked eye.

[3] Use of textual gradient feedback is intriguing. Furthermore, in 3.1.2 the authors introduce the idea of “distance between concepts” which looks novel.

Update after the rebuttal.
The authors thoroughly addressed all the weaknesses and questions I raised. I update the overall score to Accept. The change of heart comes from the updated body of measurable results from the experiments the authors have conducted during the rebuttal period. It could have been Strong Accept had the authors shown the statistical significance of the results in Table 1. Nevertheless, at its core, the paper is the first one that publicly introduces a trading bot that performs trades based on the news feed analysis by an LLM-based agent, which is very valuable for both financial and AI communities.

**Weaknesses:**

[1] In Table 2 the proposed method performs better than B&H only for TSLA and worse - for 4 other stocks: MSFT, NFLX and COIN. This result is inconclusive with respect to the question of whether the proposed method generalizes at all.

[2] Even though Figure 3 shows a clear advantage of the proposed method relative to the competitors, it is not convincing that this advantage does not come from the cherry-picking of the stocks in the “example” portfolio: TSLA, MSFT, and PFE. A more rigorous study is needed, for example to measure the average score of all possible picks of 3 stocks out of 5 available.

On line 304 the authors claim to ablate CVR, however in Table 3 for the single stock there are no numbers for w/o CVR.

[3] Combining a multi-agent setup, reinforcement Learning, CVR and convex optimization (at Manager) looks like an overly sophisticated setup. It is not clear which parts of the bundle produce how much of the impact if any.

Specifically, the Memory module consists of working memory, procedural memory, and episodic memory. The ablations for these three types of memory are not provided.

[4] The reproducibility is limited since the authors do not provide the code of their solution.
“Justification: We provide detailed experiment results.” is not on point since this clause is intended to address the reproducibility of the results by reviewer and potential future readers.

[5] Yang 2024 “Large Language Models as Optimizers” could have been cited as the first paper that proposes textual optimization by LLMs.

**Questions:**

[1] What is the difference between CVR and OPRO introduced by Yang 2024 (Large Language Models as Optimizers)?

[2] Is RL optimization performed simultaneously with CVR or one after another?

[3] On lines 708-709 you state “To mimic learning rate effects, we measure the overlapping percentage between conceptualized investment insights from consecutive iterations”. Where can I find the details of the implementation of this algorithm?

[4] On lines 222-223 the Action module in the trading scenario selects between long, short and neutral, however the volume is not specified. How is it supposed to work?

[5] In Table 1 why are there no numbers for AMZN? Also why are the best results not highlighted for stocks other than TSLA?

[6] How would you address the problem of manual labor to create specialized initial prompts for the large variety of stocks listed in Appendix J?

**Limitations:**

The limitations are well discussed.

---

> ### Author Rebuttal · Authors · 2024-08-07
>
> We appreciate the reviewer's feedback. In response to the identified weaknesses and limitations:
>
> **[W1: Experimental results] We have updated the experimental results to include other three stocks.**
>
> We have updated the experimental results to include other three stocks including AAPL, NIO and AMZN. And the new results still suggest that our model still significantly outperforms the comparative models regarding to key metrics, Cumulative Return (CR) and Sharpe Ratio (SR), as depicted in **Table A**.
>
> | Categories | Models | TSLA  | | AMZN |  | NIO  |  | AAPL |  | GOOG |  | COIN | | NFLX | |
> |------------|----------|---------------|-----------|---------------|-----------|---------------|-----------|---------------|-----------|---------------|-----------|---------------|-----------|---------------|-----------|
> | | | CR% | SR | CR% | SR | CR% | SR | CR% | SR | CR% | SR | CR% | SR | CR% | SR |
> | Market | B&H | 6.425 | 0.145 | 1.914 | 0.067|-77.210| -1.449 | 22.315 | 1.107 | 22.420 | 0.891 | -21.756 | -0.311 | 0.621 | 1.925 |
> | Our Model  | **FIN-CON** |**82.871**|**1.972**|**24.964**|**0.906**|**17.461**|**0.335**|**27.352**|**1.597**|**25.077**|**1.052**|**57.045**|**0.825**|**0.741**| **2.368**|
> | LM-Based   |GA|16.535 | 0.391| -5.515|-0.195| -3.176|-1.574| 5.694|0.372| -0.0151| -0.0192|19.271| 0.277| 0.466| 1.638|
> | | FIN-GPT  | 1.549| 0.044| -29.811| -1.805| -4.959| -0.121| 20.321| 1.161| 0.207| 0.822| -99.553|-1.807| 0.168| 0.655 |
> | | FIN-MEM  | 34.624| 1.552| -18.126| -0.776| -48.437| -1.180| 12.396| 0.994| 0.311| 0.018| 0.811| 0.017| -0.091| -0.420|
> | | FIN-AGENT| 11.960 | 0.271| -24.704| -1.496| 0.933| 0.051| 20.757| 1.041| -7.440 | -1.024| -5.971| -0.106 |0.661| 2.092|
> | DRL-based  | A2C| -35.644 | -0.805| -12.676| -0.447| -91.190| -1.728|13.781| 0.683| 8.562| 0.340|NA |NA |-0.107| -0.333|
> | | PPO | 1.409| 0.032| 3.863| 0.137| -72.119| -1.352| 14.041| 0.704 | 2.434| -0.097|NA| NA| -0.380| -1.188|
> | | DQN | -1.296| 0.029| 11.171| 0.398| -35.419| -0.662| 21.125| 1.048 | 20.690| 0.822 | NA| NA | 0.169 | 0.528|
>
> Table A: Comparative analysis of trading agent systems on single asset task: FINCON outperforms on key metrics (Cumulative Returns (CRs) and Sharpe Ratios (SRs)) across multiple stocks. _**As the COIN first IPOed in 2021, the RL algorithms failed to converge to a stable result with limited data. Thus, We marked the metrics as 'NA'. Note: Due to the space limit, we only include the values of primary metrics.**_
>
> **[W2: Stock selection and portfolio construction/ Ablation about CVR] We have clarified the criteria to select the stock and added experimental results for another portfolio. An ablation study for the CVR mechanism has also been added.]**
>
> 1. Other than simply enumerating all combinations of 3 tickers out of 5 available, our stock selection agent within FINCON constructs the portfolio based on the classic financial principle of diversification from a pool of 42 tickers (number determined by data APIs accessible to us), which achieved by constraining the statistical correlation between the candidates in this pool. Fixing a stock as the target, we found the tickers maintain the most diverse historical return patterns compared to it to form the portfolio.At the same time, we eliminated the combination with tickers has less than 1000 news records in the running period.
>    Additionally, we had the stock selection agent execute another round of portfolio construction with AMZN as the target, resulting in a new portfolio: AMZN, GM, and LLY. FINCON again exhibited significantly better performance over comparative models in terms of CR and SR, further illustrating its robustness in portfolio management tasks (**Table B** below).
>    While we currently use FINCON to trade compact portfolios (three symbols) due to time and budget constraints, its promising performance suggests a potential for managing larger portfolios.
>
> | Model | SR | CR |
> |-------|----|----|
> | FINCON | 1.501 | 37.4 |
> | Equal-weighted | 1.048 | 18.797 |
> | FinRL - A2C | 0.846 | 15.710 |
> | Markowitz MV | 0.654 | 12.983 |
>
> Table B: Key performance metric comparison among all management strategies for a portfolio consisting of (AMZN, GM, LLY). All baselines are kept the same as the ones in the paper.
>
> 2. In response to the ablation study about CVR, we have updated the results in **Table C**, as well as in [W3] for a more detailed discussion. In the table, we present the results with and without CVR updates for both single stock trading and portfolio management tasks. The results demonstrate that implementing the CVR mechanism significantly improves decision-making quality across various tasks and market conditions.
>
> |Task|Asset|MarketTrend|Models|CR%|SR|MDD%|
> |----------------------|------------------|------------------|---------|---------|--------|---------|
> |SingleStock|GOOG|GeneralBullish|w/CVR|28.972|1.233|16.990|
> ||||w/oCVR|-11.944|-0.496|29.309|
> ||NIO|GeneralBearish|w/CVR|7.981|0.157|40.647|
> ||||w/oCVR|-17.956|-0.356|55.688|
> |PortfolioManagement|(TSLA,MSFT,PFE)|Mixed|w/CVR|121.018|3.435|16.288|
> ||||w/oCVR|20.677|0.987|23.975|
>
> Table C: Key metrics FINCON with vs. without implementing Conceptual Verbal Reinforcement（CVR）/ investment belief updates for over-trajectory risk control. The performance of FINCON with the implementation of CVaR won a leading performance in both single-asset trading and portfolio management tasks.

---

> ### Author Response · Authors · 2024-08-07
> **Rebuttal Part 2**
>
> **[W3: Contribution of each FINCON component/ Ablation of memory module.] We have added an ablation study to assess the effectiveness of FINCON's episodic memory design.**
>
> 1. To clarify the contribution of each part, we provide our detailed motivation as follows:
>   - Multi-agent setup: We employ a unique Manager-Analyst hierarchical multi-agent structure, where each agent is responsible for a specific task. This synthesized approach enables collaborative tackling of sequential decision-making challenges in complex financial markets. The hierarchical communication structure enhances efficiency by eliminating redundant peer-to-peer interactions, thereby optimizing resource utilization.
>
>   - Reinforcement learning: The actor-critic structure in classical RL serves as an inspiration for our system design, though FinCon adapts its core principles rather than applying them directly. The actor component in our system is tasked with selecting policies. The critic component in our system does not estimate value functions as in traditional RL,it is designed to reflect on and learn from past actions by evaluating their outcomes based on feedback from the environment.
>
>   - Conceptual Verbal Reinforcement (CVR): The CVR helps to update manager's investment beliefs over multiple training episodes. The ablation study in **[W3.2]** below further illustrates its contribution to FINCON's performance.
>
>   - Convex optimization: Convex optimization is employed exclusively to determine asset allocation in portfolio management tasks, and we leverage this method for its proven reliability and effectiveness in portfolio management. Our work is pioneering in its integration of LLMs for managing portfolios, marking the first instance of such an application in the field.
>
>   These mechanisms work in concert to ensure FINCON's high-quality decision-making capability from different perspectives. Furthermore, we believe the setup of FINCON maintains novelty from two fronts: **Empirical novelty**: FINCON addresses the complexity of volatile market dynamics and achieves state-of-the-art performance in various practical financial decision-making tasks. Moreover, it is the first financial language agent system to support portfolio management functionality.
> **Technical novelty**: FINCON introduces the unique CVR mechanism for multi-agent system risk control. This represents a novel development beyond verbal reinforcement, designed specifically to synthesize hierarchical multi-agent collaboration.
>
> 2. Our memory module is composed of working memory, procedural memory, and episodic memory. *While the first two components support essential functions for each agent's basic operations, the episodic memory refers to the updated investment beliefs through FINCON's CVR mechanism, continuously enhancing its decision-making capabilities over episodes.* Therefore, conducting an ablation study on episodic memory is more meaningful. Our experimental outcomes are summarized in the same **Table C in W2**. Given the same choice of training and test periods as Table A and B, the results demonstrate that CRs and SRs have been substantially enhanced after iterating over four training episodes of investment belief updates. This conclusion holds true for both single stock trading and portfolio management tasks, and remains robust across various market conditions (bearish, bullish, and mixed).
>
> **[W4: Reproducibility] We have shared the source code of FinCon with standard protocol through AC.**
>
> Please don't hesitate to request it from the AC.
>
> **[W5: Citation] We will add this paper as part of our related work and citation list.**
>
> We agree the significant position of this paper in the field of textual optimization. We will incorporate it in the related work. And, the paper will also be properly cited in our reference list.
>
> **[Q1: CVR vs. OPRO] Differences Between CVR and OPRO are explained below.**
>
> 1. OPRO includes the entire optimization trajectory in meta-prompts, while CVR selects segments of consistent profit or loss.
> 2. Unlike OPRO, which only specifies the upgrade direction, CVR mimics the learning rate using the overlapping rate between conceptualized insights.
> 3. OPRO handles deterministic problems like linear regression and the traveling salesman problem, as well as manually annotated datasets (e.g., GSM8K [1], BBH [2]). In contrast, CVR addresses stochastic optimization in financial investment, managing the environment's randomness and complexity.
>
> [1] Cobbe, Karl, et al. "Training verifiers to solve math word problems." arXiv preprint arXiv:2110.14168 (2021).
>
> [2] Suzgun, Mirac, et al. "Challenging big-bench tasks and whether chain-of-thought can solve them." arXiv preprint arXiv:2210.09261 (2022).

---

> ### Author Response · Authors · 2024-08-07
> **Rebuttal Part 3**
>
> **[Q2: Optimization approach] We use an RL-inspired Actor-Critic structure to dynamically update manager agent's belief other than implementing a classic RL.**
>
> We do not employ RL optimization directly. Our work models financial trading tasks as a POMDP and addresses the optimization problem using a textual gradient approach. We incorporate the Actor-Critic (AC) structure from traditional reinforcement learning due to its broad applicability for designing intelligent agents. However, our CVR algorithm, which integrates textual gradient descent, the AC structure, and quantitative risk management, substantially differs from the traditional AC RL algorithm. Additionally, unlike traditional RL algorithms, we do not modify the intrinsic parameters of the LLMs.
>
> **[Q3: Implementation details of CVR] We have provided more details about the CVR percentage overlap across iterations. For more technical details, please request our submitted code from the AC.**
>
> In the training stage, the CVR algorithm evaluates and adjusts optimization process across training episodes by analyzing conceptualized investment insights. Overlapping percentages of insights between episodes act like a learning rate, guiding necessary adjustments: A high overlap suggests minor tweaks are sufficient, while a low overlap signals the need for more significant changes to improve performance. In our experiment, a 50% overlap between the 1st and 2nd episodes indicates significant changes are needed, while subsequent increases to 62.5% and 72.5% suggest progressively finer adjustments. This pattern shows how the algorithm adapts and refines strategies, achieving stability and improved performance through iterative learning.
>
> **[Q4:Trading volume integration] Trading volume is factored into each decision-making process, as elaborated below.**
>
> In our system, stock selection agents analyze multimodal market data to curate a stock pool based on statistical correlations between stock returns. The manager agent assigns long, short, and neutral positions to each stock, which act as constraints for the mean-variance optimization performed daily to determine portfolio weights. These weights are then linearly scaled to define target positions (scaled between 0 and 1). Finally, our back-testing system computes the number of shares to buy or sell for each asset based on its allocated buying power and current price. It is to be noticed that, though experimenting on a compact portfolio due to budget limitation, our working mechanism can be easily generalized to larger-size portfolios and is the first to employ LLM agents for managing a portfolio, marking a significant innovation in financial decision-making.
>
> **[Q5: Experimental results] We updated results with requested information.**
>
> Please refer to [W1] for detailed discussions. Please see the updated results in **Table A in [W1]**.
>
> **[Q6:Manual labor] Our prompt engineering utilizes automated generation, not solely manual labor.**
>
> The creation of ticker-specialized prompts for the profile module (Appendix J) employs an automated process. We developed a uniform prompt template that is automatically adapted for different stock tickers, significantly reducing manual effort. With approximately 3,000 actively traded US stocks, this approach is manageable with our current resources. Future implementations will incorporate advanced automated prompt-generation techniques, further minimizing manual input and enhancing prompt customization efficiency.

---

> > ### Comment · Reviewer_pGXR · 2024-08-12
> >
> > I highly appreciate your remarkably refined answers.
> >
> > W1. Thanks for the updated table of results. Whereas the updated table on its own could look convincing, I reckon several inconsistencies as below:
> > 1. The column for MSFT that was not performing well has been removed.
> > 2. “Our model” numbers for TSLA, NFLX and COIN are different in the rebuttal Table A compared to the manuscript Table 1. What is the reason for this? If it is re-running the experiments, then how can you explain such a significant variance in the evaluation metrics?
> > 3. “B&H” numbers for the same symbols in point 2 have changed as well. If for “Our model” the variation could be explained by training instability, “B&H” should be a fixed number.
> >
> > W2.1. Thank you for running the numbers for an alternative portfolio. Observing consistent performance albeit on just 2 data points is somewhat convincing.
> > W2.2-5. Well acknowledged.
> > Q1-Q6 Well acknowledged, particularly Q3, Q4 and Q6.
> >
> > I intend to revise the decision to the accepted levels. However, I would appreciate a clarification on the updated W1.

---

> > > ### Author Response · Authors · 2024-08-13
> > > **Follow-up Replies to Reviewer pGXR - Part II**
> > >
> > > 2. *More iterations of CVR and more precise in-trajectory risk control mechanism(Value at Risk [VaR])*
> > >
> > > This extension of the training and testing periods enabled us to recalibrate our model more effectively. We conducted additional ablation studies to identify the feature settings that notably enhance FINCON's performance.
> > >
> > > We found when incorporating more iterations for investment belief updates during the extended warm-up period—from two episodes in the original manuscript to four episodes. This contributes to the improvement of the trading outcomes, as shown in Table A below. The improved results stem from the acquisition of more subtle professional experience, corroborated by the learning rate enhancements detailed in our response to Q3. Accordingly, there is an increment in trading action overlapping along with the growth of the number of episodes: the overlap from the first to the second episode was 46.939%, from the second to the third 71.429%, and from the third to the fourth 81.633% (reported for TSLA, consistent with Q3).
> > >
> > > Moreover, as detailed in Section 3.1.2 of the original manuscript, we eventually decided to use a 1% threshold to trigger the VAR drop and issue an in-trajectory risk alert. This adjustment enables the agent to maintain a cautious stance toward market volatilities, adopting a risk-averse investment approach during significant market downturns yet swiftly reverting to active investment strategies following minor market fluctuations.
> > >
> > > With these investigations on features for FINCON’s risk-control component, we better demonstrate the effectiveness and robustness of our framework, despite that the longer test period means including more trading days. We will modify and include these details in our experimental settings for further revision.
> > >
> > > **[Q3: B&H numbers] -  We recalibrate it since the training and testing time period is updated.**
> > >
> > > With the updated extended testing period, we recalibrated the Buy & Hold (B&H) numbers correspondingly.
> > >
> > > ***Acknowledge:*** We have carefully made the corrections and will include the revised tables (Updated Table A) in our next revision.
> > > We will release all our code along with a well-documented README file to ensure the reproducibility of our results.

---

> ### Author Response · Authors · 2024-08-09
> **Follow-up from the authors**
>
> Thank you again for your feedback.
>
> We have added the necessary material to the author rebuttal and official comments to address the points you raised. Given that we only have three reviewers, each reviewer's score significantly impacts the overall assessment of our work. We believe that the current overall assessment does not adequately reflect the contribution of our work. Therefore, we kindly request you reconsider your score. Thank you again for your time and effort in reviewing our paper.

---

> ### Author Response · Authors · 2024-08-13
> **Follow-up Replies to Reviewer pGXR - Part 1**
>
> **[Q1： MSFT results] -  Sure!: We provided MSFT result in below.**
>
> The reason we did not add the MSFT is that we missed this stock in the updated result. Here are the updated results for MSFT. For further explanation on performance, please refer to our answer to Q2.
>
> |Categories|Models|MSFT||
> |--------|--------|--------|---------|
> |||CR%|SR|
> |Market|B&H|34.487|1.489|
> |OurModel|FINCON|**34.802**|**1.665**|
> |LLMBased|GA|-31.673|-1.374|
> ||FINGPT|20.603|1.235|
> ||FINMEM|-17.802|-0.979|
> ||FINAGENT|-27.386|-1.199|
> |DRL-Based|A2C|24.574|1.087|
> ||PPO|-7.938|-0.350|
> ||DQN|30.19|1.337|
>
> Although FINCON marginally outperforms the buy-and-hold strategy in a bullish market condition, where a naive buy strategy would be advantageous, our model gains a clear advantage over other autonomous trading systems.
>
> **[Q2: Clarification on our model results]- We have updated the experiment with extended training and test periods.**
>
> The variation in evaluation metrics for TSLA, NFLX, and COIN between Table A in the rebuttal and Table 1 in the original manuscript is not due to chance but because of the following two reasons:
>
> 1. *Extended data collection:*
>
> Initially, our training period spanned from January 2022 to August 2022, and we used testing data through April 25, 2023, as shown in Figure 3 of the original manuscript. As part of this ongoing project, we have continued to expand our data collection to include the most recent news data, updating our dataset through June 10, 2023, during the rebuttal period. This expansion has enabled us to extend the training period from January 2022 to September 2022. This longer warm-up period enabled our manager agents to receive more extensive updates on investment beliefs via CVR. Additionally, we retain a longer and more up-to-date test period (from the beginning of October 2022 to June 10th, 2023) to examine our agent performance. Our rebuttal results, including all baselines, are conducted in these same training and testing time periods.
>
> Additionally, we would like to make further clarifications for our submitted paper and first rebuttal reply: Cumulative returns (CR%) should be reported in Table 1 of the original manuscript in percentage. We input them as decimals for MSFT, AMZN, NFLX, and COIN as decimals by dismissal. This error also affected the CRs for NFLX in Table A of W1 in our rebuttal reply. We have fixed this issue and included the correct results in the following Updated Table A.
>
> |Categories|Models|TSLA||AMZN||NIO||AAPL||GOOG||COIN||NFLX||MSFT||
> |------------|----------|---------------|-----------|---------------|-----------|---------------|-----------|---------------|-----------|---------------|-----------|---------------|-----------|---------------|-----------|---------------|------------|
> |||CR%|SR|CR%|SR|CR%|SR|CR%|SR|CR%|SR|CR%|SR|CR%|SR|CR%|SR|
> |Market|B&H|6.425|0.145|1.914|0.067|-77.210|-1.449|22.315|1.107|22.420|0.891|-21.756|-0.311|62.181|1.925|34.487|1.489|
> |OurModel|FINCON|**82.871**|**1.972**|**24.964**|**0.906**|**17.461**|**0.335**|**27.352**|**1.597**|**25.077**|**1.052**|**57.045**|**0.825**|**74.082**|**2.368**|**34.802**|**1.665**|
> |LM-Based|GA|16.535|0.391|-5.515|-0.195|-3.176|-1.574|5.694|0.372|-0.0151|-0.0192|19.271|0.277|46.613|1.638|-31.673|-1.374|
> ||FINGPT|1.549|0.044|-29.811|-1.805|-4.959|-0.121|20.321|1.161|0.207|0.822|-99.553|-1.807|16.767|0.655|20.603|1.235|
> ||FINMEM|34.624|1.552|-18.126|-0.776|-48.437|-1.180|12.396|0.994|0.311|0.018|0.811|0.017|-9.135|-0.420|-17.802|-0.979|
> ||FINAGENT|11.960|0.271|-24.704|-1.496|0.933|0.051|20.757|1.041|-7.440|-1.024|-5.971|-0.106|66.145|2.092|-27.386|-1.199|
> |DRL-based|A2C|-35.644|-0.805|-12.676|-0.447|-91.190|-1.728|13.781|0.683|8.562|0.340|NA|NA|-10.677|-0.333|24.574|1.087|
> ||PPO|1.409|0.032|3.863|0.137|-72.119|-1.352|14.041|0.704|2.434|-0.097|NA|NA|-37.987|-1.188|-7.938|-0.350|
> ||DQN|-1.296|0.029|11.171|0.398|-35.419|-0.662|21.125|1.048|20.690|0.822|NA|NA|16.911|0.528|30.191|1.337|
>
> Updated Table A: Comparative analysis of trading agent systems on single asset task: FINCON outperforms on key metrics (Cumulative Returns (CRs) and Sharpe Ratios (SRs)) across multiple stocks. ***As the COIN first IPOed in 2021, the RL algorithms failed to converge to a stable result with limited data. Thus, We marked the metrics as 'NA'. Note: Due to the space limit, we only include the values of primary metrics.***

---

> > ### Comment · Reviewer_pGXR · 2024-08-13
> >
> > Thank you for addressing the reasons for the puzzling number in the original Table 1. In general, I, as a reviewer, expect the original manuscript to display the final, verified results of the work. Regarding the extended training and evaluation data, the response is accepted. Nevertheless, again, the submitted manuscript is expected to already contain convincing results, whereas in this case, only after the review and during the rebuttal process have you presented acceptable results. This is somewhat confusing.

---

> ### Comment · Reviewer_pGXR · 2024-08-13
>
> Another thing that I’d like to bring up is about your answer to “7. Experiment Statistical Significance” in the NeurIPS checklist. Speaking about Table 1, the confidence intervals are not provided, thus the answer to the questionnaire must be No. Looking at how the CR numbers vary between the original manuscript and the updated table, the CR numbers exhibit significant variation. Even the new test period size is small and can hardly be increased for the time series problems. A convincing analysis of the distribution of CR outcomes could have been done either with re-running the experiments multiple times with significant LLM temperature, or by re-running on different slices of your total data. For the latter, the example could be:
>
> Fold 1: 0-60% train, 60-90% test
>
> Fold 2: 1-61% train, 61-91% test
>
> …
>
> Fold 10: 10-70% train, 70-100% test.
>
> Either of these will produce the standard deviation of CR outcomes needed to estimate the statistical significance of the results.
>
> Anyway, even without checking the statistical significance, the current results are good. I plan to update my judgment to Accept.

---

> ### Author Response · Authors · 2024-08-13
> **Another follow-up from the authors'**
>
> We deeply appreciate your positive feedback on our submission and your willingness to consider updating the rating to acceptance level.
>
> Thank you for your insightful review and the suggestions you’ve made, which have improved the manuscript.
>
> Your point about the original vs. final manuscript is understandable: the way we see, the reviews prompted us to dig deeper on some of the questions raised, and we think inclusion of what we found in the revision is worthwhile. Your suggestion about the confidence intervals is a good one, and we will rerun our experiments based on your suggested settings and incorporate these metrics in the further revision. And we will also update the checklist.

---

> ### Author Response · Authors · 2024-08-13
>
> As the discussion deadline approaches, shall we kindly ask that you adjust the rating score to reflect your consideration of acceptance? We appreciate your support. Thanks again.

---

### Official Review · Reviewer_bNfR · 2024-07-04

**Soundness:** 3
**Presentation:** 3
**Contribution:** 3
**Rating:** 8
**Confidence:** 2

**Summary:**

The research introduces FINCON, an LLM-based multi-agent framework designed for a variety of financial tasks. FINCON is inspired by effective organizational structures in real-world investment firms and employs a manager-analyst communication hierarchy. Experimental evaluations show that FINCON’s risk control mechanism effectively mitigates investment risks and enhances trading performance. The hierarchical communication structure and risk control components improve decision quality, streamline information flow, and reduce overheads.

**Strengths:**

- The introduction of FINCON presents a novel approach by integrating Large Language Models (LLMs) into a multi-agent framework specifically designed for financial decision-making tasks.
- The dual-level risk control component that includes a self-critiquing mechanism to update systematic investment beliefs is a unique addition.
- The paper is written clearly, with well-structured sections that guide the reader through the problem, methodology, experiments, and conclusions. Each section logically follows from the previous one, making the overall argument easy to follow.
- The paper’s findings contribute significantly to the literature, offering new insights into the applications of LLMs in financial decision-making and advancing the state-of-the-art in this interdisciplinary field.

**Weaknesses:**

The inner workings of the multi-agent interactions and risk control mechanisms may not be fully transparent, which can raise trust issues among users and stakeholders who rely on the model for high-stakes financial decisions.

**Questions:**

N/A

**Limitations:**

While the model shows strong generalization capabilities in various tasks, the scalability of the model to even larger datasets, more diverse financial instruments, or real-time applications is not fully explored. The computational demands and potential bottlenecks in real-time processing need to be addressed.

---

> ### Author Rebuttal · Authors · 2024-08-07
>
> We thank the reviewer for your appreciation of our work, particularly regarding the innovative nature and efficacy of the FINCON framework. The reviewer's concerns about FINCON's weaknesses and limitations are carefully explained and addressed below.
>
> **[Weakness: Transparency] The multi-agent interactions are transparent, and the risk-control mechanism is fully accessible.**
>
> 1. The multi-agent interactions are conveyed and recorded by natural language, which is completely accessible to users. This fosters trust compared to other autonomous trading systems. This makes them suitable for monitoring and intervention if necessary. Additionally, the memories stored in the memory module can be queried in a structured form for review when needed.
>
> 2. The FINCON system features a dual-level risk control mechanism designed to enhance decision-making quality in volatile market environments while ensuring full transparency: (1) In-trajectory risk control, which monitors for sudden market risks during an ongoing episode. This mechanism is activated when there is a decrease in Conditional Value at Risk (CVaR), signaling that recent trading decisions might have pushed Profit and Loss (PnL) values into the worst 1% of outcomes, thus indicating high-risk conditions; and (2) Across-trajectory risk control, which refines the manager agent's investment strategies over multiple episodes. This is done by optimizing the integration of analyst-provided information for specific trading objectives. Both levels of risk control are operationalized through natural language prompt templates directed at the FINCON manager agent. The in-trajectory risk alerts are conveyed through real-time LLM prompts, while the across-trajectory adjustments are implemented by periodically updating the manager's profile prompts with new investment beliefs every two episodes. This structure ensures transparency of its internal mechanism and allows users to understand and potentially tailor the risk management strategies employed.
>
> However, the intrinsic mechanisms of LLM are still under active research [1]. We will add a discussion about the limitation in the future revision.
>
> [1] Luo, H., & Specia, L. (2024). From understanding to utilization: A survey on explainability for large language models. arXiv preprint arXiv:2401.12874.
>
> **[Limitation: Scalability and computational demand] Scalability and computational demand will not be a bottleneck in our case.**
>
> We appreciate the reviewer's inquiry. Our work is the first to use LLM agents for portfolio management, representing a significant innovation in AI-driven financial decision-making. While real-time processing may seem time-consuming, we operate at a daily trading frequency, a common lower-frequency approach in real markets. In our case, each decision-making step around 20 seconds to 1 minutes using an agent backed by LLM, such as GPT-4 Turbo, which is well-suited for daily trading. Given the comparison between the required trading frequency and our current average response time, FINCON demonstrates significant potential to manage larger-scale data streams and portfolios effectively, as long as the data is accessible. In our future research, we plan to incrementally test the model on larger and more complex datasets, and we also intend to incorporate a wider variety of financial instruments, such as options, futures and swaptions.

---

> > ### Comment · Reviewer_bNfR · 2024-08-10
> >
> > Thank you for your thoughtful reply on the topics of transparency, risk-control, and scalability. While I lack expertise in portfolio management or quantitative trading, I was impressed by the innovative use of large language models in a multi-agent system for financial decision-making, as demonstrated by FINCON.

---

> > > ### Author Response · Authors · 2024-08-10
> > > **Authors' response**
> > >
> > > Thanks a lot for your response.

---

### Official Review · Reviewer_rfBX · 2024-07-13

**Soundness:** 3
**Presentation:** 3
**Contribution:** 3
**Rating:** 5
**Confidence:** 2

**Summary:**

The study introduces FINCON, a large language model (LLM)-based multi-agent framework designed to improve financial decision-making,  where it utilizes a manager-analyst communication hierarchy to enhance the synthesis of multi-source information and optimize decision-making outcomes through a risk-control component and conceptual verbal reinforcement.

**Strengths:**

1. Integrating a manager-analyst hierarchical structure with LLMs to simulate the decision-making processes in financial settings is useful.

2. The study provides a strong empirical performance.

**Weaknesses:**

The implications of applying such systems in real-world financial markets may need to be discussed, including potential ethical concerns and the impact on market dynamics.

**Questions:**

1. Does FINCON demonstrate robustness across multiple financial decision-making tasks?

2. Does the risk-control component optimize analyst outcomes and manager information allocation?

3. How does FINCON perform under extreme market conditions, such as financial crises or significant market volatility?

**Limitations:**

Yes

---

> ### Author Rebuttal · Authors · 2024-08-07
>
> We thank the reviewer for recognizing the novelty and soundness of our approach. To address your mentioned weakness and questions:
>
> **[W: Ethical concerns and market impact] Clarification about Ethical Concerns & Impact on Market Dynamics**
>
>
> 1. **Regarding Data:**
>
> We fully respect the copyright of data sources and adhere to all relevant guidelines. **For public datasets,** there are no concerns regarding copyright or other restrictions. The datasets we used, such as Form 10-K, Form 10-Q, and earnings conference calls, are required filings with the SEC and are archived in publicly accessible databases. **For proprietary datasets,** we respect their copyrights and have retrieved them from valid sources with proper consent. Although we did not disclose the data to comply with copyright requirements, we have clearly specified their sources and provided the necessary code for future researchers to reproduce our work. In addition, we plan to include copyright information of our data source in Appendix B in future revisions.
>
> 2. **Regarding Market Impact**
>
> FinCon is primarily an academic exchange product and is not intended for commercial use. We will emphasize this point on our homepage, in articles, and in our code. However, if the industry is inspired by our work to design similar trading systems, we still have reason to believe that this would be beneficial for financial markets. Such a system is an automated trading system that accelerates the reflection of public information in the market, helping stock prices to converge to their true value and benefiting investors [1]. This ensures that they can trade the underlying asset at a fair price and reduce their losses due to delayed price updates. It aligns with the classical Efficient-Market Hypothesis in finance, which states that prices should immediately reflect all available information [2]. Moreover, as an assistant, the system can reduce human errors, thereby enhancing accuracy and reliability in financial operations [3]. This, in turn, decreases systemic risks in financial markets and improves the overall efficiency of financial services.
>
> By incorporating the mentioned improvements, we believe we can **fully address ethical concerns raised by reviewers**, and we think the development of such an automated trading system can **benefit the financial market.**
>
> [1] Dubey, R. K., Babu, A. S., Jha, R. R., & Varma, U. (2022). Algorithmic trading efficiency and its impact on market-quality. Asia-Pacific Financial Markets, 29(3), 381-409.
>
> [2] Miller, C. N., Roll, R., & Taylor, W. (1970). Efficient capital markets: A review of theory and empirical work. The Journal of Finance, 25(2), 383-417.
>
> [3] Todorović, V., Pešterac, A., & Tomić, N. (2019). The impact of automated trading systems on financial market stability. Facta Universitatis, Series: Economics and Organization, 255-268.
>
> **[Q1: Robustness] Generalization to Multiple Financial-decision Marking Tasks**
>
> We have conducted extensive experiments and ablation studies to demonstrate FINCON's robustness in decision-making tasks, particularly in single-stock trading and portfolio management. For more details, please check our latest results in **Table A, B, and C** comment box below and the **Table 1, 2, 3 and 4 in the pdf file attached with author rebuttal**. Our evaluation spans multiple assets under diverse market conditions. Notably, FINCON exhibits strong generalization capabilities in making automated investment decisions for both individual and combined financial products. This versatility indicates that FINCON's trading objectives could potentially extend to other financial decision-making tasks, including investments in cryptocurrencies, ETFs, or combinations thereof, provided relevant data is available. We will further add a discussion about FINCON's other potentials in our future revisions.
>
> **[Q2: Risk-control component working mechanism] FINCON's risk-control component directly impacts the manager's reasoning and influences analyst outcomes through the manager agent's feedback, which is informed by the present risk scenario.**
>
> FINCON provides a two-fold risk control mechanism to dynamically update part of the manager's decision-making contexts: (1) in-trajectory risk control alerts the manager agent to emerging market risks during active episodes, enabling immediate strategy adjustments; and (2) across-trajectory risk control adjusts the manager agent's investment beliefs for better weight allocation of information perspectives provided by each analyst. The belief updates focus on certain trading targets and is refined over episodes. In summary, our risk-control mechanism directly and timely shapes the manager agent's critical investment-related contexts in the decision-making process. The manager agent, considering these contexts, identifies critical memory insights that facilitate its decisions and informs analyst agents accordingly. The analyst agents then reinforce these memories in their memory system by increasing their ranking scores, ensuring that information considered useful for decisions has a higher chance of being selected for future decision-making and a slower decay rate from the system. Through this process, the risk-control component optimizes the workflows for both manager and analyst agents as a synthesized system.

---

> > ### Comment · Reviewer_rfBX · 2024-08-09
> >
> > I appreciate the authors' response. Although I am not an expert in financial decision-making, I find the paper's idea interesting. Based on my current understanding, I hold a somewhat positive view of the paper

---

> > > ### Author Response · Authors · 2024-08-09
> > > **Authors' response**
> > >
> > > Thank you for your positive comment.

---

> ### Author Response · Authors · 2024-08-07
> **[Q3: Extreme market conditions] We have added experimental results to demonstrate FINCON's robust performance during periods of significant market volatility.**
>
> We collected performance metrics for both single stock trading (TSLA) and portfolio management (combination of TSLA, MSFT, and PFE) tasks, focusing on Cumulative Returns (CRs) and Sharpe Ratios (SRs). The data covers a training period from 2022-01-17 to 2022-03-31 and a test period from 2022-04-01 to 2022-10-15. We selected this time range because the VIX (CBOE Volatility Index) maintained a high level (averaging above 20) during this period, indicating greater market volatility than usual.
>
> Below, we summarize the top 3 best-performing models for single stock trading in **Table D** below. FINCON is the only agent system achieving positive CRs and SRs for single stock trading tasks. For the complete results of all baselines, please refer to Table 2 in the PDF attached to the author rebuttal. For the portfolio management task, all baseline results (four benchmarks) are provided in **Table E**, in which the FINCON attained the highest values in the primary performance metrics.
>
> These empirical outcomes support FINCON's robustness during periods of significant market volatility.
>
> **Single Stock: TSLA**
>
> | Type | Model | SR | CR |
> |------|-------|----|----|
> | Our Model | FINCON | 0.695 | 22.46 |
> | Second Best | FINGPT | -0.805 | -20.035 |
> | Third Best | DQN | -1.328 | -8.452 |
>
> Table D: Top 3 best-performing models for single stock trading (TSLA) ranked by Sharpe Ratio (SR) under the high volatility condition. FinCon demonstrated the best performance for key metrics. Please refer to Table 2 in the PDF of the author rebuttal to see the full results for all baselines.
>
> **Portfolio:**
>
> | Type | Model | SR | CR |
> |------|-------|----|----|
> | Our Model | FINCON | -0.294 | -8.429 |
> | Second Best | FinRL-A2C | -1.195 | -15.932 |
> | Third Best | Equal-weighted | -1.731 | -28.008 |
> | Fourth Best | Markowitz | -1.805 | -28.996 |
>
> Table E: Key performance comparison among all portfolio management strategies under the high volatility condition. The portfolio consists of (TSLA, MSFT, PFE). All baselines are kept the same as the ones in the paper.

---

> ### Author Response · Authors · 2024-08-07
> **Experimental results for [Q1: Robustness] Generalization to Multiple Financial-decision Marking Tasks**
>
> | Categories | Models | TSLA  | | AMZN |  | NIO  |  | AAPL |  | GOOG |  | COIN | | NFLX | |
> |------------|----------|---------------|-----------|---------------|-----------|---------------|-----------|---------------|-----------|---------------|-----------|---------------|-----------|---------------|-----------|
> | | | CR% | SR | CR% | SR | CR% | SR | CR% | SR | CR% | SR | CR% | SR | CR% | SR |
> | Market | B&H | 6.425 | 0.145 | 1.914 | 0.067|-77.210| -1.449 | 22.315 | 1.107 | 22.420 | 0.891 | -21.756 | -0.311 | 0.621 | 1.925 |
> | Our Model  | **FIN-CON** |**82.871**|**1.972**|**24.964**|**0.906**|**17.461**|**0.335**|**27.352**|**1.597**|**25.077**|**1.052**|**57.045**|**0.825**|**0.741**| **2.368**|
> | LM-Based   |GA|16.535 | 0.391| -5.515|-0.195| -3.176|-1.574| 5.694|0.372| -0.0151| -0.0192|19.271| 0.277| 0.466| 1.638|
> | | FIN-GPT  | 1.549| 0.044| -29.811| -1.805| -4.959| -0.121| 20.321| 1.161| 0.207| 0.822| -99.553|-1.807| 0.168| 0.655 |
> | | FIN-MEM  | 34.624| 1.552| -18.126| -0.776| -48.437| -1.180| 12.396| 0.994| 0.311| 0.018| 0.811| 0.017| -0.091| -0.420|
> | | FIN-AGENT| 11.960 | 0.271| -24.704| -1.496| 0.933| 0.051| 20.757| 1.041| -7.440 | -1.024| -5.971| -0.106 |0.661| 2.092|
> | DRL-based  | A2C| -35.644 | -0.805| -12.676| -0.447| -91.190| -1.728|13.781| 0.683| 8.562| 0.340|NA |NA |-0.107| -0.333|
> | | PPO | 1.409| 0.032| 3.863| 0.137| -72.119| -1.352| 14.041| 0.704 | 2.434| -0.097|NA| NA| -0.380| -1.188|
> | | DQN | -1.296| 0.029| 11.171| 0.398| -35.419| -0.662| 21.125| 1.048 | 20.690| 0.822 | NA| NA | 0.169 | 0.528|
>
> Table A: Comparative analysis of trading agent systems on single asset task: FINCON outperforms on key metrics (Cumulative Returns (CRs) and Sharpe Ratios (SRs)) across multiple stocks. _**As the COIN first IPOed in 2021, the RL algorithms failed to converge to a stable result with limited data. Thus, We marked the metrics as 'NA'. Note: Due to the space limit, we only include the values of primary metrics.**_
>
> | Model | SR | CR |
> |-------|----|----|
> | FINCON | 1.501 | 37.4 |
> | Equal-weighted | 1.048 | 18.797 |
> | FinRL - A2C | 0.846 | 15.710 |
> | Markowitz MV | 0.654 | 12.983 |
>
> Table B: Key performance metric comparison among all management strategies for an additional portfolio consisting of (AMZN, GM, LLY). All baselines are kept the same as the ones in the paper.
>
> | Task                 | Asset            | Market Trend     | Models  | CR%     | SR     | MDD%    |
> |----------------------|------------------|------------------|---------|---------|--------|---------|
> | Single Stock         | GOOG             | General Bullish  | w/CVR   | 28.972  | 1.233  | 16.990  |
> |                      |                  |                  | w/o CVR | -11.944 | -0.496 | 29.309  |
> |                      | NIO              | General Bearish  | w/CVR   | 7.981   | 0.157  | 40.647  |
> |                      |                  |                  | w/o CVR | -17.956 | -0.356 | 55.688  |
> | Portfolio Management | (TSLA, MSFT, PFE)| Mixed            | w/CVR   | 121.018 | 3.435  | 16.288  |
> |                      |                  |                  | w/o CVR | 20.677  | 0.987  | 23.975  |
>
> Table C: Key metrics FINCON with vs. without implementing Conceptual Verbal Reinforcement（CVR）/ investment belief updates for over-trajectory risk control. The performance of FINCON with the implementation of CVaR won a leading performance in both single-asset trading and portfolio management tasks.

---

### Author Rebuttal · Authors · 2024-08-07

We sincerely appreciate all reviewers for their time and insightful feedback. We are glad that many reviewers found that:

**Our framework is novel and our design of risk-control component is unique.**

- [bNfR] The introduction of FINCON presents a novel approach ... for financial decision-making tasks./ The dual-level risk control component ... is a unique addition.
- [pGXR] The paper tries to apply LLM agents to financial problems which is a very novel...

**Our approach is to be technically sound. And our empirical performance to be solid.**
- [rfBX] Integrating a manager-analyst hierarchical structure ... is useful. The study provides a strong empirical performance.
- [bNfR] Technically strong paper, with novel ideas, excellent impact on at least one area...
- [pGXR] The study is well structured and encloses the comparison with 6 SOTA competitors and the B&H baseline ... Specifically Figure 3 shows impressive results visible by a naked eye.


**Our contribution to financial decision-making field is significant.**
- [bNfR]The paper’s findings contribute significantly to the literature ... in financial decision-making and advancing the state-of-the-art in this interdisciplinary field
- [pGXR] The paper tries to apply LLM agents to financial problems ... promising field of study.

Again, we would like to thank you for your recognition.
Regarding limitations, beyond minor presentational issues and factual clarifications, two primary concerns emerged:

**More experiments**

- [rfBX] asked for FINCON'S performance in the period with significant market volatility.We have conducted additional performance evaluations for both single-asset trading and portfolio management tasks during a notably volatile time period, distinct from the experimental conditions in our original paper.

- [pGXR] on single-asset trading tasks and ablation studies focusing on the memory module and Conceptual Verbal Reinforcement (CVR). We have expanded the asset range and refined our experimental set-up to deliver the requested experiment outcomes. Furthermore, we provided the requested ablation study, which can help to illustrates the effectiveness of our memory module design as well as CVR mechanism simultaneously.

_**Please check out our detailed experimental results in the attached pdf file and replies below.**_

**Further explanation of working mechanism**
- [bNfR] "The inner workings of the multi-agent interactions and risk control mechanisms may not be fully transparent..." We have included a comprehensive explanation detailing how FINCON ensures full transparency in its multi-agent interaction workflow and risk-control mechanisms.
- [pGXR] "On lines 708-709 ... Where can I find the details of the implementation of this algorithm?”; "On lines 222-223 ... How is it supposed to work?" We have provided detailed explanations for all requested aspects of FINCON's operational mechanisms in our responses to the reviewers. If our paper is luckily accepted, we will incorporate this additional information into the revised manuscript.

In the meantime, we have also **clarified the ethical concern** from reviewer [rfBX]:  "The implications of applying such systems in real-world financial markets may need to be discussed, including potential ethical concerns and the impact on market dynamics." We've provided a detailed clarification addressing potential ethical concerns. We demonstrate that, other than undermining the market, implementing automated trading systems like FINCON can actually benefit the financial market dynamics. Our explanation includes comprehensive rationales supporting this conclusion.

Last, we would like to respectfully note that **our code was submitted to the Area Chair through an anonymized link via an author rebuttal, in full compliance with the review process guidelines.** If our submission is accepted, we commit to making our code publicly available. This release aims to provide valuable resources to researchers and developers in the financial technology and language agent design communities, fostering broader advancements in these fields. **Moreover, we would like to claim that our code is intended solely for academic and educational purposes. The code does not constitute financial, legal, or investment advice and is not intended to be a basis for any decision-making.**

---

### Author Response · Authors · 2024-08-14
**To SAC/AC (and Thanks to All Reviewers).**

Dear Area Chair,

We are blessed to receive an engaging author-reviewer discussion and glad that all three of the reviewers are on the positive side of the scale. Reviewer [rfBX] and [bNfR] hold their original **positive scores (5 and 8)**, and reviewer [pGXR] agreed to increase the rating from 3 to **an acceptance level.**


Again, as noted in our General Response below, we are grateful that most reviewers found our work, A LLM multi-agent system with conceptual verbal reinforcement for multiple financial decision to be:
-    **novel in framework design**([bNfR] and [pGXR])
-    **technically sound in method** ([rfBX], [bNfR], [pGXR])
-    **solid in empirical evaluation** ([bNfR], [pGXR])
-    **contributed significantly to financial decision-making field** ([bNfR]).


In response to the questions and concerns raised by the reviewers—[rfBX], [bNfR], and [pGXR]—we have been asked to:

- Conduct more ablation studies to demonstrate the **necessity of our risk-control component**.
- Perform additional experiments to prove the **effectiveness of our framework under extreme market conditions** [[Q3: Extreme market conditions](https://openreview.net/forum?id=dG1HwKMYbC&noteId=w8InKmaChX)] and to illustrate **our method's robustness across a broader spectrum of single-stock trading tasks and portfolio management tasks** [[Q1: Robustness](https://openreview.net/forum?id=dG1HwKMYbC&noteId=5mwNeuSFl2)].
- Provide detailed technical explanations and expand on some domain-specific backgrounds, highlighting the **scalability** of our system and explaining its **benefits to the financial market and investors**.[[W:Scalability and computational demand](https://openreview.net/forum?id=dG1HwKMYbC&noteId=tt3Gh7E5BP), [Limitation:Ethical concerns and market impact](https://openreview.net/forum?id=dG1HwKMYbC&noteId=g1JVgumRr1)]

We believe these concerns have been effectively addressed after we conducted additional experiments and engaged in thorough discussions with the specific reviewers. At the same time, we also collected many useful insights and suggestions on our work. We sincerely appreciate the thoughtful suggestions by the reviewers.

Considering our detailed rebuttal and the existing scores, we are confident that the solidness and comprehensiveness of our work align well with NeurIPS standards. While we fully respect your final decision, we would like to respectfully highlight the potential broader impact of this work could have on both the AI for finance and language agent system communities. Our work stands out for several pioneering aspects :

1. **We are the first to implement a language agent system to address portfolio optimization, one of the most challenging decision-making tasks in finance.**

2. **We introduce an innovative textual optimization method called conceptual verbal reinforcement, which facilitates efficient hierarchical collaboration between Manager and Analyst agents.**

3. **We uniquely craft the agents' personas with a dual-level risk control component, the importance of which in enhancing decision quality is robustly supported by our ablation studies.**

And these apsects have been recognized by all reviewers through the discussion.



Thank you for your time and efforts in organizing and overseeing the paper review process. We would like to bring to your attention an issue concerning the review from [pXGR]. ***In [pXGR] last communication, [pXGR] agreed to update the judgment to 'Accept,([Comment:pXGR](https://openreview.net/forum?id=dG1HwKMYbC&noteId=ykqXsb1Tfs)) , yet this change has not been reflected in [pXGR] official rating. We are concerned this may be due to an oversight or an unusual circumstance, as [pXGR] intention was clearly stated in the last response. Additionally, we have addressed all the questions [pXGR] raised during the rebuttal period with detailed experiments and explanations. Could you please check with reviewer [pXGR] to confirm his or her current position if possible?***


Sincerely,

Paper 13270 Authors

---

### Decision · Program_Chairs · 2024-09-25

**Decision:**

Accept (poster)

**Comment:**

This paper proposes a comprehensive agent-based approach to using LLMs to manage financial decision making, specifically, single-security and whole portfolio trading/investment decisions. The breakdown of tasks is nominally based on the organizational breakdown of investment organizations (e.g., with specialist analysts that focus on specific types of (multi-modal) information, a single portfolio manager than makes weighting and long/short/neutral decisions for individual securities, a risk mgmt agent). This decomposition, along with the use of textual reinforcement, blended with standard financial investment techniques provide the major contribution of the paper.

On the positive side, the paper provides a comprehensive use of LLMs, with, arguably, a fairly intuitive and useful approach to decomposition. The paper is fairly clearly written (though see below) and the results (at least as described in the rebuttal) seem fairly compelling.

On the negative side (and I agree with all of these points on my reading of the paper):

1. The reviewers express some concerns about the overall transparency of the approach (not to “users” but to researchers/readers of this paper). This is a complex system, and basically impossible to describe in a 9pp NeurIPS paper. The authors do a reasonable job of giving the high-level picture in the main body of the paper, but there are many, many details are pushed into a lengthy—total PDF of 40pp—appendix (including precisely how the LLM agents are prompted, how prompt optimization is conducted, almost all of the precise details about the experiments, etc.). To the extent that a NeurIPS paper should be reasonably self-contained, I think this paper fails (which is unavoidable because of the topic/complexity).

2. The results in the paper do not actually look convincing. E.g., Table 1, on my reading, shows that FINCON performs worse than at least one other strategy—and often worse than buy-and-hold—for most of the individual securities (if lower returns are not balanced by improvements in Sharpe and drawdown metrics, this is a bad result). It seems better on portfolio mgmt (Table 2) compared to some reasonable baselines, but leads to some serious questions about the results. In fact, this lead the original reviews to provide some very low scores. However, in the rebuttal, the authors provided updated (and I believe corrected results) that were much more impressive, so much so that the reviewers changed their (aggregate) assessment significantly. The results in the rebuttal are indeed much more compelling than those in the original submission.

On the first issue, I worry if NeurIPS is the right forum for this paper; but given the potential importance of the topic, the interest it might generate, and the fact that the appendix does make important details available, I think this can be overlooked.

On the second issue: I cannot recommend acceptance of the original manuscript. But if the updated results are incorporated and explained well (and the majority of them *must* be in the main body of the paper), then I think the paper would make a nice contribution. The authors must commit to making revisions along these lines. Moreover, a number of other, smaller issues raised by the reviewers in their reviews and the discussion, should be addressed if the paper is accepted. (This includes providing some evidence of the statistical significance of the comparisons being made against the competing methods.)